# WIN: WEIGHT-DECAY-INTEGRATED NESTEROV ACCELERATION FOR ADAPTIVE GRADIENT ALGORITHMS

**Pan Zhou**[1]     **Xingyu Xie**[2,1]     **Shuicheng Yan**[1]
[1]Sea AI Lab
[2]National Key Lab of General AI, School of Intelligence Science and Technology, Peking University
{zhoupan,xyxie,yansc}@sea.com

## ABSTRACT

Training deep networks on large-scale datasets is computationally challenging. In this work, we explore the problem of "*how to accelerate adaptive gradient algorithms in a general manner*", and aim to provide practical efficiency-boosting insights. To this end, we propose an effective and general Weight-decay-Integrated Nesterov acceleration (Win) to accelerate adaptive algorithms. Taking AdamW and Adam as examples, we minimize a dynamical loss per iteration which combines the vanilla training loss and a dynamic regularizer inspired by proximal point method (PPM) to improve the convexity of the problem. To introduce Nesterov-alike-acceleration into AdamW and Adam, we respectively use the first- and second-order Taylor approximations of vanilla loss to update the variable twice. In this way, we arrive at our Win acceleration for AdamW and Adam that uses a conservative step and a reckless step to update twice and then linearly combines these two updates for acceleration. Next, we extend Win acceleration to LAMB and SGD. Our transparent acceleration derivation could provide insights for other accelerated methods and their integration into adaptive algorithms. Besides, we prove the convergence of Win-accelerated adaptive algorithms and justify their convergence superiority over their non-accelerated counterparts by taking AdamW and Adam as examples. Experimental results testify to the faster convergence speed and superior performance of our Win-accelerated AdamW, Adam, LAMB and SGD over their non-accelerated counterparts on vision classification tasks and language modeling tasks with both CNN and Transformer backbones. We hope Win shall be a default acceleration option for popular optimizers in deep learning community to improve the training efficiency. Code will be released at https://github.com/sail-sg/win.

## 1 INTRODUCTION

Deep neural networks (DNNs) are effective to model realistic data and have been successfully applied to many applications, *e.g.* image classification (He et al., 2016) and speech recognition (Sainath et al., 2013). Typically, their training models can be formulated as a nonconvex problem:

$$\min_{\boldsymbol{z} \in \mathbb{R}^d} F(\boldsymbol{z}) := \mathbb{E}_{\boldsymbol{\zeta} \sim \mathcal{D}}[f(\boldsymbol{z}, \boldsymbol{\zeta})] + \frac{\lambda}{2} \|\boldsymbol{z}\|_2^2, \tag{1}$$

where $\boldsymbol{z} \in \mathbb{R}^d$ is the model parameter; sample $\boldsymbol{\zeta}$ is drawn from a data distribution $\mathcal{D}$; the loss $f$ is differentiable; $\lambda$ is a constant. Though many algorithms, *e.g.* gradient descent (Cauchy et al., 1847) and variance-reduced algorithms (Rie Johnson, 2013), can solve problem (1), SGD (Robbins & Monro, 1951) uses the compositional structure in (1) to efficiently estimate gradient via minibatch data, and has become a dominant algorithm to train DNNs in practice because of its higher efficiency and effectiveness. However, on sparse data or ill-conditioned problems, SGD suffers from slow convergence speed (Kingma & Ba, 2014), as it scales the gradient uniformly in all parameter coordinate and ignores the problem properties on each coordinate. To resolve this issue, recent work has proposed a variety of adaptive methods, *e.g.* Adam (Kingma & Ba, 2014) and AdamW (Loshchilov & Hutter, 2018), that scale each gradient coordinate according to the current geometry curvature of the loss $F(\boldsymbol{z})$. This coordinate-wise scaling greatly accelerates the optimization convergence and helps them, *e.g.* Adam and AdamW, become more popular in DNN training, especially for transformers.

Unfortunately, along with the increasing scale of both datasets and models, efficient DNN training even with SGD or adaptive algorithms has become very challenging. In this work, we are particularly interested in the problem of "*how to accelerate the convergence of adaptive algorithms in a general manner*" because of their dominant popularity across many DNNs. Heavy ball acceleration (Polyak, 1964) and Nesterov acceleration (Nesterov, 2003) are widely used in SGD but are rarely studied in adaptive algorithms. Among the very few, NAdam (Dozat, 2016) simplifies Nesterov acceleration to estimate the first moment of gradient in Adam while totally ignoring the second-order moments, which is not exact Nesterov acceleration and may not inherit its full acceleration merit.

**Contributions:** In this work, based on a recent Nesterov-type acceleration formulation (Nesterov et al., 2018) and proximal point method (PPM) (Moreau, 1965), we propose a new *Weight-decay-Integrated Nesterov acceleration* (Win for short) to accelerate adaptive algorithms, and also further analyze the convergence of Win-accelerated adaptive algorithms to justify their convergence superiority by taking AdamW and Adam as examples. Our main contributions are highlighted below.

Firstly, we use PPM to rigorously derive our Win acceleration for accelerating adaptive algorithms. By taking AdamW and Adam as examples, at the $k$-iteration, we follow PPM spirit and minimize a dynamically regularized loss $F(z) + \frac{1}{2\eta_k} \|z - x_k\|^2_{\sqrt{v_k + \nu}}$ with the second-order gradient moment $v_k$ and the stabilizing constant $\nu$ in AdamW and Adam. Then to introduce Nesterov-alike acceleration and also make the problem solvable iteratively, we respectively approximate $F(z)$ by its first- and second-order Taylor expansions to update the variable $z$ twice while always fixing the above dynamic regularization and also an extra regularizer $\frac{1}{2\eta_k} \|z\|^2_{\sqrt{v_k + \nu}}$ induced by the weight decay in AdamW. As a result, we arrive at our Win acceleration, a Nesterov-alike acceleration, for AdamW and Adam that uses a conservative step and a reckless step to update twice and then linearly combines these two updates for acceleration. Then we extend this Win acceleration to LAMB (You et al., 2019) and SGD. The above acceleration derivation is transparent and general which could motivate other accelerations and provide examples to introduce other accelerations into adaptive algorithms.

Secondly, we prove the convergence of our Win-accelerated AdamW and Adam. For both, to find an $\epsilon$-approximate first-order stationary point, their stochastic gradient complexity is $\mathcal{O}\left(\frac{c_\infty^{2.5}}{\nu^{1.25}\epsilon^4}\right)$ and matches the lower bound $\Omega(\frac{1}{\epsilon^4})$ in (Arjevani et al., 2019; 2020) (up to constant factors) under the same conditions, where $c_\infty$ upper bounds the $\ell_\infty$ norm of stochastic gradient. Moreover, this complexity improves a factor $\mathcal{O}(\frac{d}{c_\infty^{0.5}})$ over the complexity $\mathcal{O}(\frac{c_\infty^2 d\sigma^2 L}{\nu^{1.25}\epsilon^4})$ of Adam-type optimizers in (Zhou et al., 2018; Guo et al., 2021), *e.g.* Adam, AdaGrad (Duchi et al., 2011), AdaBound (Luo et al., 2018), since network parameter dimension $d$ is often much larger than $c_\infty^{0.5}$, especially for over-parameterized networks. Indeed, Win-accelerated Adam and AdamW also enjoy superior complexity than other Adam variants, *e.g.* Adabelief (Zhuang et al., 2020) with compelxity $\mathcal{O}(\frac{c_2^6}{\nu^2\epsilon^4})$, especially on over-parameterized networks, where $c_2$ is the maximum $\ell_2$-norm of stochastic gradient.

Finally, experimental results on both vision classification tasks and language modeling tasks show that our Win-accelerated algorithms, i.e. accelerated AdamW, Adam, LAMB and SGD, can accelerate the convergence speed and also improve the performance of their corresponding non-accelerated counterparts by a remarkable margin on both CNN and transformer architectures. All these results show the strong compatibility, generalization and superiority of our acceleration technique.

## 2 RELATED WORK

In the context of deep learning, when considering efficiency and generalization, one often prefers to adopt SGD and adaptive gradient algorithms, *e.g.* Adam, instead of other algorithms, *e.g.* variance-reduced algorithms (Rie Johnson, 2013), to solve problem (1). But, in practice and theory, adaptive algorithms often suffer from inferior generalization performance than SGD (Zhou et al., 2020a;b). To solve this issue, AdamW (Loshchilov & Hutter, 2018) proposes a decoupled weight decay which introduces an $\ell_2$-alike regularization into Adam to decay network weight iteratively, and its effectiveness is widely validated on ViTs (Touvron et al., 2021) and CNNs (Touvron et al., 2021). Later, LAMB (You et al., 2019) scales the update in AdamW to the weight magnitude for avoiding too large or small update, but suffers from unsatisfactory performance on small batch. In this work, we aim to design a general acceleration to accelerate these adaptive algorithms.

Heavy-ball acceleration (Polyak, 1964) and Nesterov acceleration (Nesterov, 2003) are two classical acceleration techniques, and their effectiveness in SGD is well testified. Later, NAdam (Dozat,

2016) integrates Nesterov acceleration into the first-order gradient moment estimation but ignores the second-order gradient moments which harms the acceleration effect. Some works (Anil et al., 2022; 2020) also explore Nesterov acceleration for second-order algorithms, *e.g.* shampoo (Gupta et al., 2018). Recently, for full gradient decent algorithm, a new general Nesterov-type acceleration (Nesterov et al., 2018) directly interpolates two variables to look ahead for correction, and is more flexible than vanilla Nesterov acceleration (Nesterov, 2003) which interpolates the variable and gradient. See discussion in Sec. 3.2. Here we use proximal point method to introduce this new acceleration into adaptive algorithms by a rigorous and transparent derivation and necessary tailors.

# 3 WEIGHT-DECAY-INTEGRATED NESTEROV ACCELERATION

To accelerate full gradient descent algorithm, given a full gradient $\nabla F(z_k)$ of problem (1) at the $k$-th iteration, Nesterov-type acceleration (Nesterov et al., 2018) generally uses a conservative step $\eta_k$ and a reckless step $\bar{\eta}_k$ to update two sequences $x_{k+1}$ and $y_{k+1}$ respectively, and then linearly combines them to update the variable $z_{k+1}$ of the problem. Similar formulations are also observed and proved in recent works, *e.g.* (Allen-Zhu & Orecchia, 2014; Bansal & Gupta, 2019; Ahn & Sra, 2022). In general, their acceleration formulation can be formally formulated as

$$x_{k+1} = x_k - \eta_k \nabla F(z_k), \quad y_{k+1} = z_k - \bar{\eta}_k \nabla F(z_k), \quad z_{k+1} = \rho_k x_{k+1} + (1 - \rho_k) y_{k+1}. \quad (2)$$

This acceleration enjoys provably faster convergence rate for full gradient descent method on convex problems (Nesterov et al., 2018), and is then empirically validated in many convex and nonconvex cases, *e.g.* (Wilson et al., 2017; Nado et al., 2021). Despite its effectiveness, such acceleration is rarely explored in adaptive gradient algorithms, especially for network training. In deterministic optimization setting, another widely used optimization-stabilizing and acceleration approach is proximal point method (PPM) (Moreau, 1965; Rockafellar, 1976). At the $k$-th iteration, PPM optimizes an $\ell_2$-regularized loss $F(z) + \frac{1}{2\eta_k}\|z - z_{k-1}\|_2^2$ instead of the vanilla loss $F(z)$. This small change enhances the convexity of the problem, accelerating and also stabilizing optimization process (Kim et al., 2022; Zhou et al., 2021c). To make the $\ell_2$-regularized problem solvable iteratively, PPM approximates the loss $F(z)$ by its first- or second-order Taylor expansion so that each iteration has a close-form solution (see below). At below, we borrow the idea in PPM to induce a *Weight-decay-Integrated Nesterov acceleration* (Win) for adaptive algorithms by using AdamW and Adam as examples in Sec. 3.1, and then extend this acceleration technique to LAMB and SGD in Sec. 3.2.

## 3.1 WIN-ACCELERATED ADAMW AND ADAM

To begin with, following most adaptive gradient algorithms, *e.g.* Adam and AdamW, we estimate the first- and second-order moments $m_k$ and $v_k$ of gradient as follows:

$$g_k = \frac{1}{b} \sum_{i=1}^{b} \nabla f(z_k; \zeta_i), \quad m_k = (1 - \beta_1) m_{k-1} + \beta_1 g_k, \quad v_k = (1 - \beta_2) v_{k-1} + \beta_2 g_k^2, \quad (3)$$

where $g_k$ is the average gradient on a minibatch data of size $b$, $\beta_1 \in [0, 1]$ and $\beta_2 \in [0, 1]$. For the initialization, we set $m_0 = g_0$, $v_0 = g_0^2$. For brevity, with a small scaler $\nu > 0$, we define

$$s_k = \sqrt{v_k + \nu}, \qquad u_k = m_k / \sqrt{v_k + \nu}. \quad (4)$$

Then following the spirit of PPM, at the $k$-th iteration, we minimize a regularized loss $F(x) + \frac{1}{2\eta_k}\|x - x_k\|_{s_k}^2$, where $\|x\|_{s_k} = \sqrt{\langle x, s_k * x \rangle}$ with an element-wise product operation $*$. Here we use the regularizer $\|x - x_k\|_{s_k}^2$ instead of the $\ell_2$-regularization $\|x - x_k\|_2^2$, since 1) this new regularization can induce adaptive algorithms as shown below Eqn. (5), and 2) it increases the convexity of the problem and further considers different sharpness property of each coordinate because of different elements in $s_k$, accelerating convergence. To make the problem solvable iteratively, we approximate the vanilla loss $F(z)$ by its first-order Taylor expansion at the point $z_k$ and update $x_{k+1}$ as

$$x_{k+1} = \operatorname{argmin}_x F(z_k) + \langle m_k, x - z_k \rangle + \frac{1}{2\eta_k}\|x - x_k\|_{s_k}^2 + \frac{\lambda_k}{2}\|x\|_{s_k}^2 = \frac{1}{1 + \lambda_k \eta_k}(x_k - \eta_k u_k), \quad (5)$$

where $m_k$ is used to approximate the full gradient $\nabla F(z_k)$. We add a small regularization $\frac{\lambda_k}{2}\|x\|_{s_k}^2$, since 1) it can largely improve the generalization performance in practice (Loshchilov & Hutter, 2018; Touvron et al., 2021); 2) it allows us to derive Adam ($\lambda_k = 0$) and AdamW ($\lambda_k > 0$). Here $\lambda_k$ can be fixed as a constant or evolved along iteration number $k$. But in practice, a evolving $\lambda_k$ often enjoys better performance than a fixed one (Caron et al., 2021; Zhou et al., 2022). When $\lambda_k = 0$, the updating (5) becomes the exact Adam. If $\lambda_k > 0$, the updating (5) can approximate the updating rule $x_{k+1} = (1 - \lambda_k \eta_k) x_k - \eta_k u_k$ of AdamW. This is because consider $\lambda_k \eta_k$ is small in

practice, we can approximate $(1+\lambda_k\eta_k)^{-1} = 1 - \lambda_k\eta_k + \mathcal{O}(\lambda_k^2\eta_k^2)$ and thus $\frac{1}{1+\lambda_k\eta_k}(\boldsymbol{x}_k - \eta_k\boldsymbol{u}_k) = [1 - \lambda_k\eta_k + \mathcal{O}(\lambda_k^2\eta_k^2)]\boldsymbol{x}_k - [\eta_k - \mathcal{O}(\lambda_k\eta_k^2) + \mathcal{O}(\lambda_k^3\eta_k^3)]\boldsymbol{u}_k$ which becomes AdamW by ignoring the ignorable terms $\mathcal{O}(\eta_k^2)$ or $\mathcal{O}(\eta_k^3)$. This is also one reason that we adopt the regularizer $\|\boldsymbol{x} - \boldsymbol{x}_k\|_{\boldsymbol{s}_k}^2$ in (5) instead of the $\ell_2$-regularization in PPM, since we can flexibly derive Adam and AdamW.

Similarly, we minimize a regularized loss $F(\boldsymbol{z}) + \frac{1}{2\eta_k}\|\boldsymbol{z} - \boldsymbol{x}_{t+1}\|_{\boldsymbol{s}_k}^2$ again, and further approximate $F(\boldsymbol{z})$ by its second-order approximation $F(\boldsymbol{z}_k) + \langle\boldsymbol{m}_k, \boldsymbol{z} - \boldsymbol{z}_k\rangle + \frac{1}{2\bar{\eta}_k}\|\boldsymbol{z} - \boldsymbol{z}_k\|_{\boldsymbol{s}_k}^2$:

$$\boldsymbol{z}_{k+1} = \arg\min_{\boldsymbol{z}} F(\boldsymbol{z}_k) + \langle\boldsymbol{m}_k, \boldsymbol{z} - \boldsymbol{z}_k\rangle + \frac{1}{2\bar{\eta}_k}\|\boldsymbol{z} - \boldsymbol{z}_k\|_{\boldsymbol{s}_k}^2 + \frac{1}{2\eta_k}\|\boldsymbol{z} - \boldsymbol{x}_{k+1}\|_{\boldsymbol{s}_k}^2 + \frac{\lambda_k}{2}\|\boldsymbol{z}\|_{\boldsymbol{s}_k}^2 \quad (6)$$
$$= \bar{\eta}_k\tau_k\boldsymbol{x}_{k+1} + \eta_k\tau_k\big(\boldsymbol{z}_k - \bar{\eta}_k\boldsymbol{u}_k\big),$$

where $\tau_k = \frac{1}{\eta_k+\bar{\eta}_k+\lambda_k\eta_k\bar{\eta}_k}$, $\boldsymbol{m}_k$ is used to approximate $\nabla F(\boldsymbol{x}_k)$ as guaranteed by Theorem 1 in Sec. 4, $\bar{\eta}_k$ approximates the inverse of the local smoothness parameter of $F(\boldsymbol{z})$ around $\boldsymbol{z}_k$. Here we use a regularizer $\|\boldsymbol{z} - \boldsymbol{x}_{k+1}\|_{\boldsymbol{s}_k}^2$ with the latest update $\boldsymbol{x}_{k+1}$ instead of $\boldsymbol{x}_k$ as an anchor point, since the latest update $\boldsymbol{x}_{k+1}$ could often provide better regularization for the concurrent optimization.

Now we have used PPM to rigorously derive our Win-accelerated AdamW and Adam in Eqns. (3), (5) and (6). For more clarity, we summarize their algorithmic steps in Algorithm 1 in which we omit the bias-correction term for simplicity. When $\lambda = 0$, it is Win-accelerated Adam; if $\lambda > 0$, it gives Win-accelerated AdamW. Generally, AdamW can greatly improve the generalization performance of Adam by simply adding a weight decay (i.e. the regularizer $\frac{\lambda}{2}\|\cdot\|_{\boldsymbol{s}_k}^2$) into Adam as observed in many works, *e.g.* (Loshchilov & Hutter, 2018; Touvron et al., 2021). Our Win-acceleration is quit simple and efficient, since our accelerated AdamW/Adam only adds one extra simple algorithmic step, i.e. the seventh step in Algorithm 1, on vanilla AdamW/Adam, and brings negligible extra computational overhead into vanilla optimizer, e.g. about $2\% \sim 5\%$ extra training time per iteration on AdamW evaluated on ViT-small and ViT-base. Moreover, for the only extra hyper-parameter, the reckless step $\bar{\eta}_k$, in Algorithm 1 over AdamW/Adam, we always set it $2\times$ larger than the conservative step $\eta_k$ for all iterations, i.e. $\bar{\eta}_k = 2\eta_k$, working well in our all experiments.

Now we discuss the relations between Nesterov-type acceleration (2) and our Win acceleration (6). For comparison, we introduce a virtual sequence $\boldsymbol{y}_{k+1} = \boldsymbol{z}_k - \bar{\eta}_k\boldsymbol{u}_k$ in Win, and rewrite (6) as

$$\boldsymbol{x}_{k+1} = (1 + \lambda_k\eta_k)^{-1}(\boldsymbol{x}_k - \eta_k\boldsymbol{u}_k), \quad \boldsymbol{y}_{k+1} = \boldsymbol{z}_k - \bar{\eta}_k\boldsymbol{u}_k, \quad \boldsymbol{z}_{k+1} = \bar{\eta}_k\tau_k\boldsymbol{x}_{k+1} + \eta_k\tau_k\boldsymbol{y}_{k+1}, \quad (7)$$

where $\boldsymbol{u}_k$ is defined in (4). By comparing Nesterov-type acceleration (2) with our Win acceleration (7), one can observe some similarity and also differences as well. For similarity, both methods use a conservative step $\eta_k$ and a reckless step $\bar{\eta}_k$ to update $\boldsymbol{x}_{k+1}$ and $\boldsymbol{y}_{k+1}$ respectively, and then linearly combine $\boldsymbol{x}_{k+1}$ and $\boldsymbol{y}_{k+1}$ to obtain $\boldsymbol{z}_{k+1}$. For the differences, the first one is that Win has a weight-decay-alike factor $\frac{1}{1+\lambda_k\eta_k}$ in (7) which slightly decays the variable $\boldsymbol{x}_k$ like AdamW and also the update $\boldsymbol{u}_k$, while Nesterov acceleration does not have. Note, weight decay can greatly benefit generalization in practice as shown in many works, *e.g.* (Loshchilov & Hutter, 2018; Touvron et al., 2021; Liu et al., 2021). Another difference is that for almost all acceleration techniques, including Nesterov-type acceleration (2), the sum of their linear combination factors (*e.g.* $\rho_k$ and $1 - \rho_k$ in (2)) is always one. In contrast, in Eqn. (7), Win uses $\bar{\eta}_k\tau_k + \eta_k\tau_k = 1 - \frac{\lambda_k\eta_k\bar{\eta}_k}{\eta_k+\bar{\eta}_k+\lambda_k\eta_k\bar{\eta}_k} < 1$ when $\lambda_k > 0$, which further gives a second weigh decay. Since these two differences are caused by the weight decay, we call our acceleration "weight-decay-integrated Nesterov acceleration" (Win for short).

## 3.2 EXTENSION TO LAMB AND SGD

Here we generalize Win acceleration to LAMB (You et al., 2019) and SGD (Robbins & Monro, 1951). For LAMB, it scales the update $\boldsymbol{u}_k$ of AdamW in Eqn. (4) so that $\boldsymbol{u}_k$ is at the same magnitude of the network weight $\boldsymbol{x}_k$. That is, it changes the update rule $\boldsymbol{x}_{k+1} = (1 - \lambda_k\eta_k)\boldsymbol{x}_k - \eta_k\boldsymbol{m}_k/\boldsymbol{s}_k$ in AdamW to $\boldsymbol{x}_{k+1} = \boldsymbol{x}_k - \eta_k\frac{\|\boldsymbol{x}_k\|_2}{\|\boldsymbol{r}_k+\lambda_k\boldsymbol{x}_k\|_2}(\boldsymbol{r}_k + \lambda_k\boldsymbol{x}_k)$ where $\boldsymbol{r}_k = \boldsymbol{m}_k/\boldsymbol{s}_k$. This modification is to avoid too large or small update, improving optimization efficiency. To extend Win acceleration to LAMB, we inherit this scaling spirit, and scale the update $\boldsymbol{u}_k$ in (4) to the following one:

$$\boldsymbol{u}_k = (\|\boldsymbol{x}_k\|_2/\|\boldsymbol{r}_k + \lambda_k\boldsymbol{x}_k\|_2) \cdot (\boldsymbol{r}_k + \lambda_k\boldsymbol{x}_k). \quad (8)$$

We scale $\boldsymbol{m}_k/\boldsymbol{s}_k$ instead of $(\boldsymbol{m}_k/\boldsymbol{s}_k + \lambda_k\boldsymbol{x}_k)$ in LAMB, as our scaling can be repeatedly used to update our two sequences $\boldsymbol{x}_k$ and $\boldsymbol{z}_k$. Next, we can respectively follow Eqn. (5) and (6) to update the two sequences $\boldsymbol{x}_k$ and $\boldsymbol{z}_k$. See the detailed steps of Win-accelerated LAMB in Algorithm 1, and the detailed comparison between LAMB and Win-accelerated LAMB in Appendix A.

---

**Algorithm 1: Win-Accelerated AdamW, Adam and LAMB**

---

**Input:** initialization $\boldsymbol{x}_0 = \boldsymbol{z}_0 = \boldsymbol{0}$, step size $\{(\eta_k, \bar{\eta}_k)\}_{k=0}^T$, moment parameters $\{\beta_1, \beta_2\}$.
**Output:** $(\bar{\boldsymbol{x}}, \bar{\boldsymbol{z}})$ uniformly seleted from $\{(\boldsymbol{x}_k, \boldsymbol{z}_k)\}_{k=0}^T$.

1 **while** $k < T$ **do**

2      $\boldsymbol{g}_k = \frac{1}{b} \sum_{i=1}^b \nabla f(\boldsymbol{z}_k; \boldsymbol{\zeta}_i)$

3      $\boldsymbol{m}_k = (1 - \beta_1) \boldsymbol{m}_{k-1} + \beta_1 \boldsymbol{g}_k$                             /\* $\boldsymbol{m}_0 = \boldsymbol{g}_0$ \*/

4      $\boldsymbol{v}_k = (1 - \beta_2) \boldsymbol{v}_{k-1} + \beta_2 \boldsymbol{g}_k^2$                              /\* $\boldsymbol{v}_0 = \boldsymbol{g}_0^2$ \*/

5      $\boldsymbol{u}_k = \frac{\boldsymbol{m}_k}{\sqrt{\boldsymbol{v}_k} + \nu}$ for AdamW and Adam, $\boldsymbol{u}_k = \frac{\|\boldsymbol{x}_k\|_2}{\|\boldsymbol{m}_k / \sqrt{\boldsymbol{v}_k + \nu} + \lambda_k \boldsymbol{x}_k\|_2} \left( \frac{\boldsymbol{m}_k}{\sqrt{\boldsymbol{v}_k} + \nu} + \lambda_k \boldsymbol{x}_k \right)$ for LAMB

6      $\boldsymbol{x}_{k+1} = \frac{1}{1 + \lambda_k \eta_k} (\boldsymbol{x}_k - \eta_k \boldsymbol{u}_k)$

7      $\boldsymbol{z}_{k+1} = \bar{\eta}_k \tau_k \boldsymbol{x}_{k+1} + \eta_k \tau_k (\boldsymbol{z}_k - \bar{\eta}_k \boldsymbol{u}_k)$ with $\tau_k = \frac{1}{\eta_k + \bar{\eta}_k + \lambda_k \eta_k \bar{\eta}_k}$

8 **end while**

---

For SGD, applying Win acceleration to it is quite direct. Specifically, the only algorithmic difference between SGD and AdamW on the $\ell_2$-regularized problems is that SGD has no second-order moment $\boldsymbol{v}_k$ while AdamW has. So we can borrow the acceleration framework of AdamW in Sec. 3.1 to accelerate SGD by setting $\boldsymbol{s}_k = \boldsymbol{1} \in \mathbb{R}^d$ in Eqn. (4), (5) and (6), and obtain Win-accelerated SGD:

$$\boldsymbol{m}_k = \beta_1 \boldsymbol{m}_{k-1} + \beta_1' \boldsymbol{g}_k, \ \ \boldsymbol{x}_{k+1} = \frac{1}{1 + \lambda_k \eta_k} (\boldsymbol{x}_k - \eta_k \boldsymbol{m}_k), \ \ \boldsymbol{z}_{k+1} = \bar{\eta}_k \tau_k \boldsymbol{x}_{k+1} + \eta_k \tau_k (\boldsymbol{z}_k - \bar{\eta}_k \boldsymbol{m}_k), \ \ (9)$$

where $\beta_1' \in [0, 1]$ is dampening parameter. Here we slightly modify the moment $\boldsymbol{m}_k$ to accord with the one used in Nesterov-accelerated SGD (*e.g.* SGD-M in Pytorch) whose updating steps are

$$\boldsymbol{m}_k = \beta_1 \boldsymbol{m}_{k-1} + \beta_1' (\boldsymbol{g}_k + \lambda_k \boldsymbol{x}_k), \quad \boldsymbol{x}_{k+1} = (1 - \lambda_k \eta_k) \boldsymbol{x}_k - \eta_k (\boldsymbol{g}_k + \beta_2 \boldsymbol{m}_k). \quad (10)$$

By comparing Win-accelerated SGD and SGD-M in (10), one can find their big differences mainly caused by their different acceleration strategies and ways to handle weight decay. Win-accelerated SGD is derived from PPM and a recently proposed acceleration (2), while SGD-M modifies another previous Nesterov-type acceleration (Nesterov, 2003) (of formulation $\boldsymbol{m}_k = \beta_1 \boldsymbol{m}_{k-1} - \frac{\eta_k}{b} \sum_{i=1}^b \nabla f(\boldsymbol{x}_k + \eta_k \boldsymbol{m}_{k-1}; \boldsymbol{\zeta}_i)$ and $\boldsymbol{x}_{k+1} = \boldsymbol{x}_k + \boldsymbol{m}_k$) to better train networks. See more mechanisms of previous Nesterov acceleration and (10) in (Sutskever et al., 2013; Bengio et al., 2013).

## 4 CONVERGENCE ANALYSIS

Here we investigate the convergence performance of Win-accelerated algorithms by taking accelerated AdamW, Adam and SGD as examples, as these algorithms are more preferably used in deep learning field. Moreover, since we aim to accelerate deep network training which is highly nonconvex problems, we focus on analyzing nonconvex problems to accord with the practical setting.

For analysis, we follow previous optimization works, *e.g.* (Kingma & Ba, 2014; Reddi et al., 2019; Duchi et al., 2011; Zhou et al., 2020b; 2021a;b; Xie et al., 2022), to introduce necessary assumptions.

**Assumption 1** (*L*-smoothness). *We say a function $f(\boldsymbol{z}, \cdot)$ to be L-smooth w.r.t. $\boldsymbol{z}$, if for $\forall \boldsymbol{z}_1, \boldsymbol{z}_2$ and $\forall \boldsymbol{\zeta} \sim \mathcal{D}$, we have $\|\nabla f(\boldsymbol{z}_1, \boldsymbol{\zeta}) - \nabla f(\boldsymbol{z}_2, \boldsymbol{\zeta})\|_2 \leq L \|\boldsymbol{z}_1 - \boldsymbol{z}_2\|_2$ with a universal constant L.*

**Assumption 2** (Unbiased and bounded gradient estimation). *The gradient estimation $\boldsymbol{g}_k$ is unbiased, i.e. for $\forall k$, $\mathbb{E}[\boldsymbol{g}_k] = \nabla F(\boldsymbol{z}_k)$, and its magnitude and variance are bounded, namely, for $\forall k$, $\|\boldsymbol{g}_k\|_\infty \leq c_\infty$ and $\mathbb{E}[\|\nabla F(\boldsymbol{z}_k) - \boldsymbol{g}_k\|_2] \leq \sigma$ with two universal constants $c_\infty$ and $\sigma$.*

Next, we first define a dynamic function $F_k(\boldsymbol{z})$ at the $k$-th iteration which is real loss minimized by our algorithms. It combines the vanilla loss $F(\boldsymbol{z})$ in (1) and a dynamic regularization $\frac{\lambda_k}{2} \|\boldsymbol{z}\|_{\boldsymbol{s}_k}^2$:

$$F_k(\boldsymbol{z}) = F(\boldsymbol{z}) + \frac{\lambda_k}{2} \|\boldsymbol{z}\|_{\boldsymbol{s}_k}^2 = \mathbb{E}_{\boldsymbol{\zeta}}[f(\boldsymbol{z}; \boldsymbol{\zeta})] + \frac{\lambda_k}{2} \|\boldsymbol{z}\|_{\boldsymbol{s}_k}^2, \quad (11)$$

where $\boldsymbol{s}_k$ is given in (4). To obtain (11), following PPM spirit and Eqn. (5), one can approximate $F(\boldsymbol{z})$ by its first-order Taylor expansion, and obtain Eqn. (5) with $\boldsymbol{x}$ replaced by $\boldsymbol{z}$ to update $\boldsymbol{z}_{k+1} = \frac{1}{1 + \lambda_k \eta_k} (\boldsymbol{z}_k - \eta_k \boldsymbol{m}_k / \boldsymbol{s}_k)$. Since $\lambda_k \eta_k$ is very small, one can follow the discussion below Eqn. (5) and approximate $\boldsymbol{z}_{k+1}$ as $\boldsymbol{z}_{k+1} = (1 - \lambda_k \eta_k) \boldsymbol{z}_k - \eta_k \boldsymbol{m}_k / \boldsymbol{s}_k$ which becomes the update rule of AdamW. This is the reason that our analysis on Win-accelerated AdamW involves a dynamic loss $F_k(\boldsymbol{z})$ in (11). Note, for Win-accelerated Adam ($\lambda_k = 0$), $F_k(\boldsymbol{z})$ degenerates to the vanilla loss $F(\boldsymbol{z})$.

With these assumptions, we analyze the convergence behaviors of our accelerated algorithms on general nonconvex problems, and summarize our main results in Theorem 1 with proof in Appendix E.

**Theorem 1.** *Suppose Assumptions 1 and 2 hold, and $\boldsymbol{x}_\star \in \arg\min_{\boldsymbol{x}} F(\boldsymbol{x})$. Let $\bar{\eta}_k = \gamma \eta_k, \gamma > 1, \eta_k = \eta \leq \mathcal{O}\left(\frac{\nu^{1.25} b \epsilon^2}{c^{1.5} \gamma^{2.5} \sigma^2 L}\right), \beta_1 \leq \mathcal{O}\left(\frac{\nu^{0.5} b \epsilon^2}{c \sigma^2}\right), \beta_2 \in (0,1), c = (c_\infty^2 + \nu)^{0.5}, \lambda_k = \lambda(1 - \frac{\beta_2 c_\infty^2}{\nu})^k \ (k > 0)$ and $\lambda_0 = 0$ with a constant $\lambda > 0$. Then after $T = \mathcal{O}\left(\frac{c_\infty^{2.5} \gamma^{2.5} \sigma^2 L \Delta}{\nu^{1.25} b \epsilon^4}\right)$ iterations with minibatch size $b$ and $\Delta = F(\boldsymbol{x}_0) - F(\boldsymbol{x}_\star)$, the sequence $\{(\boldsymbol{x}_k, \boldsymbol{z}_k)\}_{k=0}^T$ generated by Win-accelerated AdamW and Adam in Algorithm 1 satisfies the following four properties.*

*a) The gradient $\nabla F_k(\boldsymbol{x}_k)$ of the sequence $\{\boldsymbol{x}_k\}_{k=0}^T$ can be upper bounded by*

$$\frac{1}{T}\sum_{k=0}^{T-1} \mathbb{E}\left[\|\nabla F_k(\boldsymbol{x}_k)\|_2^2 + \frac{1}{4}\|\boldsymbol{m}_k + \lambda_k \boldsymbol{x}_k * \boldsymbol{s}_k\|_2^2\right] \leq \epsilon^2.$$

*b) The gradient moment $\boldsymbol{m}_k$ can well estimate the full gradient $\nabla F(\boldsymbol{x}_k)$ and $\nabla F(\boldsymbol{z}_k)$:*

$$\frac{1}{T}\sum_{k=0}^{T-1} \max\left\{\mathbb{E}\|\boldsymbol{m}_k - \nabla F(\boldsymbol{x}_k)\|_2^2, \mathbb{E}\|\boldsymbol{m}_k - \nabla F(\boldsymbol{z}_k)\|_2^2\right\} \leq \left(16 + \frac{1}{2c}\nu^{0.5}L\right)\epsilon^2.$$

*c) The sequence $\{\boldsymbol{x}_k, \boldsymbol{z}_k\}$ satisfies*

$$\frac{1}{T}\sum_{k=0}^{T-1}\left\{\mathbb{E}\|\boldsymbol{x}_k - \boldsymbol{x}_{k+1}\|_{\boldsymbol{s}_k}^2, \mathbb{E}\|\boldsymbol{z}_{k+1} - \boldsymbol{z}_k\|_2^2, \mathbb{E}\|\boldsymbol{z}_k - \boldsymbol{x}_k\|_2^2\right\} \leq \left\{4\eta^2\epsilon^2, \frac{\nu^{1.5}\beta_1^2\epsilon^2}{4c(1-\beta_1)^3 L^2}, \frac{\nu^{0.5}\epsilon^2}{4cL}\right\}.$$

*d) The total stochastic gradient complexity to achieve the above three properties is $\mathcal{O}\left(\frac{c_\infty^{2.5}\Delta\sigma^2 L}{\nu^{1.25}\epsilon^4}\right)$.*

Theorem 1 guarantees the convergence of Win-accelerated AdamW and Adam in Algorithm 1 on nonconvex problems. When $\lambda_k > 0$ ($\lambda_k = 0$), Algorithm 1 corresponds to Win-accelerated AdamW (Adam). For both cases, Theorem 1 holds. Theorem 1 a) shows that by running at most $T = \mathcal{O}\left(\frac{c_\infty^{2.5}\Delta\sigma^2 L}{\nu^{1.25}b\epsilon^4}\right)$ iterations, the average gradient $\frac{1}{T}\sum_{k=0}^{T-1}\mathbb{E}\left[\|\nabla F_k(\boldsymbol{x}_k)\|_2^2\right]$ is upper bounded by $\epsilon^2$, guaranteeing the algorithmic convergence. Theorem 1 b) indicates the gradient moment $\boldsymbol{m}_k$ can well estimate the full gradient $\nabla F(\boldsymbol{z}_k)$ and also $\nabla F(\boldsymbol{x}_k)$ because of their small distances, guaranteeing the good Taylor approximation used in Eqns. (5) and (6). Moreover, in Theorem 1 c), one can find that although Algorithm 1 uses a conservative step $\eta_k$ and a reckless step $\bar{\eta}_k = \gamma\eta_k$ ($\forall\gamma > 1$) to update, the two sequences $\boldsymbol{x}_{k+1}$ and $\boldsymbol{z}_{k+1}$ can converge to each other, which could be the key for the good convergence behavior of both Win-accelerated AdamW and Adam.

Now we discuss the stochastic gradient complexity of Win-accelerated Adam and AdamW. Theorem 1 d) shows that to find an $\epsilon$-approximate first-order stationary point, both Win-accelerated Adam and AdamW have the complexity $\mathcal{O}\left(\frac{c_\infty^{2.5}\sigma^2 L}{\nu^{1.25}\epsilon^4}\right)$ which matches the lower bound $\Omega\left(\frac{1}{\epsilon^4}\right)$ in (Arjevani et al., 2019; 2020) (up to constant factors) under the same Assumptions 1 and 2. Our accelerated Adam and AdamW enjoy superior complexity over Adam-type optimizers, *e.g.* Adam, AdaGrad (Duchi et al., 2011), AdaBound (Luo et al., 2018), whose previously best known complexity under the same assumptions is $\mathcal{O}\left(\frac{c_\infty^2 d\sigma^2 L}{\nu^{1.25}\epsilon^4}\right)$ in (Zhou et al., 2018; Chen et al., 2021; Guo et al., 2021). By comparison, both accelerated Adam and AdamW improve their complexity by a factor $\mathcal{O}\left(\frac{d}{c_\infty^{0.5}}\right)$, where the network parameter dimension $d$ is often much larger than $c_\infty^{0.5}$, especially for over-parameterized modern networks. Since the convergence of AdamW has not been proved yet in the literatures, here we cannot directly compare with it. Moreover, the complexity of Win-accelerated Adam and AdamW is also lower than $\mathcal{O}\left(\frac{c_2^6\sigma^2 L}{\nu^2\epsilon^4}\right)$ of Adabelief (Zhuang et al., 2020) and $\mathcal{O}\left(\frac{c_\infty^{0.5}d^{0.5}\sigma^2 L}{\nu\epsilon^4}\right)$ of RMSProp (Tijmen & Geoffrey, 2012; Zhou et al., 2018), especially on over-parameterized networks, since for a $d$-dimensional gradient, its $\ell_2$-norm upper bound $c_2$ is often much larger than the $\ell_\infty$-norm $c_\infty$ and can be $\sqrt{d}\times$ larger for worse case.

Now we discuss the convergence performance of Win-accelerated SGD in Theorem 2.

**Theorem 2.** *Suppose Assumptions 1 and 2 hold, and $\boldsymbol{x}_\star \in \arg\min_{\boldsymbol{x}} F(\boldsymbol{x})$. Let $\bar{\eta}_k = \gamma\eta_k, \gamma > 1, \eta_k = \eta \leq \mathcal{O}\left(\frac{b\epsilon^2}{c^{1.5}\gamma^{2.5}\sigma^2 L}\right), \beta_1 \leq \mathcal{O}\left(\frac{b\epsilon^2}{c\sigma^2}\right), \beta_1' = 1 - \beta_1, \lambda_k = \lambda, \lambda_0 = 0$. After $T = \mathcal{O}\left(\frac{\Delta\sigma^2 L}{b\epsilon^4}\right)$ iterations with minibatch size $b$ and $\Delta = F(\boldsymbol{x}_0) - F(\boldsymbol{x}_\star)$, the sequence $\{(\boldsymbol{x}_k, \boldsymbol{z}_k)\}_{k=0}^T$ generated by Win-accelerated SGD in (9) satisfies the four properties in Theorem 1 with $\nu = c_\infty = c = 1$ and $\boldsymbol{s}_k = \boldsymbol{1} \in \mathbb{R}^d$.*

See its proof in Appendix F. Theorem 2 also guarantees the convergence of Win-accelerated SGD. By using the hyper-parameter settings in Theorem 2, the sequence $\{(\boldsymbol{x}_k, \boldsymbol{z}_k)\}_{k=0}^T$ generated by Win-accelerated SGD satisfies the four properties in Theorem 1 with $\nu = c_\infty = c = 1$ and $\boldsymbol{s}_k = \boldsymbol{1}$. It shows the complexity $\mathcal{O}\left(\frac{L\sigma^2}{\epsilon^4}\right)$ of Win-accelerated SGD which also matches the lower bound $\Omega\left(\frac{1}{\epsilon^4}\right)$ in (Arjevani et al., 2019; 2020) (up to constant factors) under Assumptions 1 and 2.

Table 2: ImageNet top-1 accuracy (%) of ResNet50&101 whose official optimizer is LAMB due to the stronger data augmentation for better performance. ∗ is reported in (Wightman et al., 2021).

| Epoch | ResNet50 | | | | ResNet101 | | | |
|---|---|---|---|---|---|---|---|---|
| | 100 | 200 | 300 | avg. | 100 | 200 | 300 | avg. |
| SAM | 77.3 | 78.7 | 79.4 | 78.5 | 79.5 | 81.1 | 81.6 | 80.7 |
| SGD-H | 75.3 | 76.9 | 77.2 | 76.5 | 77.7 | 78.6 | 78.8 | 78.4 |
| SGD-M | 77.0 | 78.6 | 79.3 | 78.3 | 79.3 | 81.0 | 81.4 | 80.6 |
| SGD-Win | 78.0 | 79.2 | 79.7 | $79.0_{+0.7}$ | 80.1 | 81.2 | 81.6 | $81.0_{+0.4}$ |
| Adam | 76.9 | 78.4 | 78.8 | 78.1 | 78.4 | 80.2 | 80.6 | 79.7 |
| Adam-Win | 77.4 | 78.8 | 79.3 | $78.5_{+0.4}$ | 79.2 | 80.6 | 81.0 | $80.3_{+0.6}$ |
| AdamW | 77.0 | 78.9 | 79.3 | 78.4 | 78.9 | 79.9 | 80.4 | 79.7 |
| AdamW-Win | 78.0 | 79.3 | 79.9 | $79.1_{+0.7}$ | 80.2 | 81.1 | 81.3 | $80.9_{+1.2}$ |
| LAMB | 77.0 | 79.2 | 79.8∗ | 78.7 | 79.4 | 81.1 | 81.3∗ | 80.6 |
| LAMB-Win | 78.4 | 79.7 | 80.1 | $79.4_{+0.7}$ | 80.6 | 81.5 | 81.7 | $81.2_{+0.6}$ |

# 5 EXPERIMENTS

Here we evaluate our accelerated algorithms on two representative tasks, including vision classification tasks and natural language modeling tasks. For vision tasks, we test accelerated algorithms on both CNNs, *e.g.* ResNet (He et al., 2016), and vision transformers (ViTs), *e.g.* ViT (Dosovitskiy et al., 2020) and PoolFormer (Yu et al., 2021; 2022). For language modeling tasks, we use LSTM (Schmidhuber et al., 1997) and Transformer-XL (Dai et al., 2019) for evaluation.

For clarity, we call our accelerated algorithm "X-Win", where "X" denotes vanilla optimizers, *e.g.* Adam. In all experiments, we do not change model architectures and data augmentations, and only replace the default optimizer with ours. Moreover, for all experiments, our accelerated algorithms, *e.g.* AdamW-Win, always use the default optimizer-inherent hyper-parameters of the vanilla optimizers, *e.g.* first- and second-order moment parameters $\beta_1$ and $\beta_2$ in AdamW; and their reckless step $\bar{\eta}_k$ always satisfies $\bar{\eta}_k = 2\eta_k$. These settings well reduce the parameter-tuning cost of our algorithms. In the experiments, same with other optimizers, we only slightly tune other widely tuned hyper-parameters around the vanilla ones, *e.g.* step size and warm-up epochs, *etc*, which is reasonable, as our accelerated algorithms have two step sizes and the vanilla ones are not very suitable.

## 5.1 RESULTS ON VISION CLASSIFICATION TASKS

**Results on ResNet18.** Here we follow the conventional supervised training setting used in ResNets (He et al., 2016) and evaluate our accelerated algorithms on ImageNet (Fei-Fei, 2009). Due to limited space, we defer the hyper-parameter settings of the four accelerated algorithms in Table 1 into Appendix B.

Table 1 shows that our accelerated algorithms can improve the corresponding non-accelerated versions by a remarkable margin. For instance, AdamW-Win, Adam-Win and LAMB-Win respectively make 3.1%, 2.8% and 2.6% improvement over their corresponding non-accelerated coun-

Table 1: ImageNet top-1 accuracy (%) of ResNet18. ∗, † and ‡ are respectively reported in (Chen et al., 2021), (Zhuang et al., 2020) and (Liu et al., 2019).

| | | | |
|---|---|---|---|
| AdaBound | 68.1∗ | Radam | 67.7∗ |
| Nadam | 68.8 | Padam | 70.1∗ |
| Yogi | 68.2∗ | AdaBelief | 70.1† |
| SGD-H | 67.3 | Adam-M | 67.7 |
| SGD-M | 70.2∗ | Adam | 66.5‡ |
| SGD-Win | $70.7_{+0.5}$ | Adam-Win | $69.3_{+2.8}$ |
| AdamW | 67.9∗ | LAMB | 68.5 |
| AdamW-Win | $71.0_{+3.1}$ | LAMB-Win | $71.1_{+2.6}$ |

terparts, AdamW, Adam and LAMB. Moreover, SGD-Win improves SGD-H (i.e. SGD + heavy ball) by 3.4%, and also surpasses SGD-M ( Nesterov-accelerated SGD in Sec. 3.2) by 0.5%, also validating the superiority of our Win acceleration. Besides, our accelerated algorithms, i.e. SGD-Win, AdamW-Win and LAMB-Win, beat several other optimizers, *e.g.* AdaBound, Radam (Liu et al., 2019), Nadam, Padam (Chen et al., 2021), AdaBelief, Yogi (Zaheer et al., 2018), in which Nadam uses Nesterov acceleration to estimate its first-order gradient moment. Actually, LAMB-Win sets a new SoTA top-1 accuracy on ResNet18. All these results show the strong compatibility and superiority of our Win-acceleration in adaptive algorithms.

**Results on ResNet50&101.** Here we adopt the training setting in (Wightman et al., 2021) to train ResNet50&101, as this setting uses stronger data augmentation and largely improves CNNs' performance. See augmentation details and our algorithmic hyper-parameter settings in Appendix B. Here LAMB is the default optimizer because of its higher performance than other optimizers caused by the stronger augmentations (Wightman et al., 2021). All optimizers in Table 2 are under this setting.

Table 2 shows that our accelerated algorithms consistently outperform their corresponding non-accelerated version. For example, across the three training epoch settings on ResNet50 / ResNet101,

Table 3: ImageNet top-1 accuracy (%) of ViT and PoolFormer whose default optimizers are both AdamW. ∗ and ⋄ are respectively reported in (Touvron et al., 2021) and (Yu et al., 2021).

| | ViT-S | | | ViT-B | | | PoolFormer-S12 | | |
|---|---|---|---|---|---|---|---|---|---|
| Epoch | 150 | 300 | avg. | 150 | 300 | avg. | 150 | 300 | avg. |
| SGD-M | 77.4 | 79.4 | 78.4 | 79.6 | 80.0 | 79.8 | 69.7 | 74.3 | 72.0 |
| SGD-Win | 78.1 | 80.1 | $79.1_{+0.7}$ | 80.4 | 80.8 | $80.6_{+0.8}$ | 71.1 | 74.5 | $72.8_{+0.8}$ |
| Adam | 77.3 | 79.3 | 78.3 | 79.0 | 79.7 | 79.4 | 74.3 | 76.3 | 75.3 |
| Adam-Win | 78.6 | 80.2 | $79.4_{+1.1}$ | 80.0 | 80.5 | $80.3_{+0.9}$ | 75.6 | 77.1 | $76.4_{+1.1}$ |
| AdamW | 78.3 | $79.8^*$ | 79.1 | 79.5 | $81.8^*$ | 80.7 | 75.2 | $77.1^*$ | 76.2 |
| AdamW-Win | 79.3 | 81.0 | $80.2_{+1.1}$ | 81.0 | 82.3 | $81.7_{+1.0}$ | 76.7 | 77.6 | $77.2_{+1.0}$ |
| LAMB | 78.0 | 79.6 | 78.8 | 80.3 | 80.8 | 80.6 | 75.4 | 77.4 | 76.4 |
| LAMB-Win | 79.3 | 80.6 | $80.0_{+1.2}$ | 81.0 | 81.4 | $81.2_{+0.6}$ | 76.7 | 78.0 | $77.4_{+1.0}$ |

LAMB-Win always achieves remarkable improvement over the official optimizer LAMB for this training recipe. Specifically, LAMB-Win makes 0.7% average improvement over LAMB on both ResNet50 / ResNet101. For AdamW-Win and Adam-Win, they also respectively improve their counterparts by 0.7% and 0.4% on ResNet50, 1.2% and 0.6% on ResNet101. SGD-Win also makes 2.5% and 0.8% overall improvement over heavy-ball accelerated SGD (SGD-H) and Nesterov accelerated SGD (SGD-M) on ResNet50, and also has similar advantage on ResNet101. These improvements are not trivial because of the following two reasons. 1) Since the performance is already high and may approach the model limit, it is already very hard to make very large improvement. This is testified by the fact that in (Wightman et al., 2021), using LAMB to train ResNet50 for 600 epochs only gives 80.4% top-1 accuracy. In contrast, our accelerated LAMB-Win uses 300 epochs (half training cost) to achieve 80.2%. 2) By comparing the previous optimizers, including SAM, SGD-M, Adam, AdamW and LAMB, one can observe smaller accuracy gap ($\leq 0.2\%$) between the best optimizer and the runner-up. For example, on ResNet101, the SoTA optimizer, i.e. SAM, only makes 0.1% average improvement over the runner-up LAMB. All these comparisons show the non-travail improvement of our accelerated algorithms over their counterparts.

**Results on ViTs.** We follow the widely used official training setting of ViTs (Touvron et al., 2021; Yu et al., 2021). To evaluate the performance of our accelerated algorithms, we select two popular and representative ViT architures, including ViT (Dosovitskiy et al., 2020) and PoolFormer (Yu et al., 2021). See the training setting and our hyper-parameter settings in Appendix B.

We test our accelerated algorithms under different model sizes and different training epochs, and report the results in Table 3. One can find that since AdamW and LAMB use the decoupled weight decay, they enjoy better performance than SGD and Adam, which is also observed in other works, *e.g.* (Xiao et al., 2021; Nado et al., 2021). Moreover, under different training settings, our accelerated algorithms consistently outperform the corresponding non-accelerated counterparts. Specifically, compared the default AdamW optimizer on both ViT and PoolFormer, our accelerated AdamW-Win respectively makes about 1.0%, 0.9%, 1.0% average improvement under the two training epoch settings on ViT-S, ViT-B and PoolFormer-S12. For Adam-Win and LAMB-Win, one can also observe their remarkable improvements on the three ViT backbones. Moreover, our accelerated SGD-Win also outperforms the Nesterov-accelerated SGD denoted as "SGD-M" by non-trivial margins under all settings. All these results are consistent with the observations on ResNets, and they together demonstrate the advantage of our accelerated optimizers for deep network training.

**Results Analysis.** Here we investigate the convergence behaviors of our accelerated algorithms, and aim to explain their better test performance over their non-accelerated counterparts. In Fig. 1, we plot the curves of training and test losses along with the training epochs on ResNet18 and ViT-B. One can find that our accelerated algorithms, *e.g.* AdamW-Win, show much faster convergence behaviors than their non-accelerated counterparts, *e.g.* AdamW. Moreover, SGD-Win also converges faster than Nesterove-accelerated SGD, i.e. SGD-M. We also plot the curves of test accuracy in Fig. 2, showing the superior convergence speed of AdamW-Win and LAMB-Win over their non-accelerated versions. Fig. 3 in Appendix B also reveals SGD-Win and

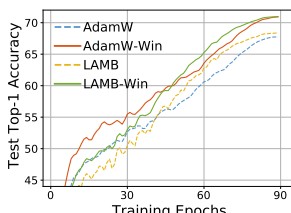

Figure 2: Test accuracy curves of AdamW-Win and LAMB-Win on ResNet18.

Adam-Win enjoy faster convergence than their non-accelerated counterparts in terms of test accuracy. So these faster convergence behaviors could contribute to our accelerated algorithms for their higher performance over non-accelerated counterparts under the same computational cost.

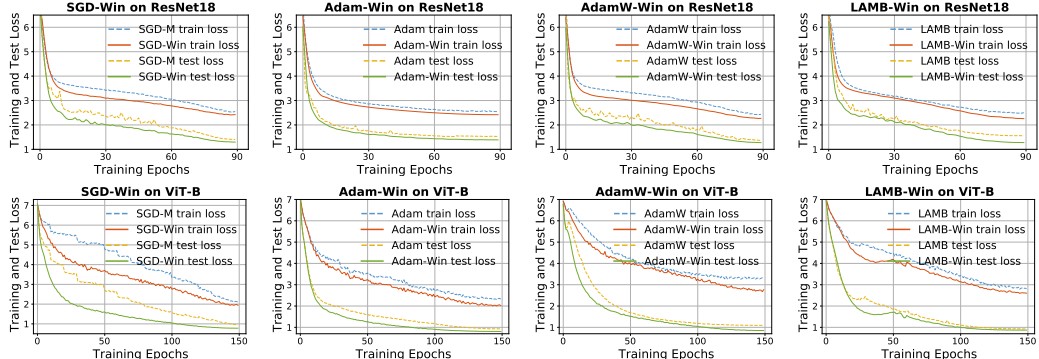

Figure 1: Visualization of training and test losses on ImageNet. In all figures, training loss is larger than test one, as training data use random augmentations, *e.g.* random crop and clip, while test data only adopt the centralization crop which eases the recognition difficulty and thus has small loss.

**Robust Analysis.** For the only extra hyper-parameter $\bar{\eta}_k$ in our accelerated algorithms over their non-accelerated counterparts, in experiments, we always set $\bar{\eta}_k = \gamma \eta_k$, where $\gamma = 2$. Here we investigate the effects of $\gamma$ to the accelerated algo-

Table 4: Effects of $\gamma$ to top-1 accuracy (%) of AdamW-Win and LAMB-Win on ResNet50.

| $\gamma$ | 1.5 | 2 | 3 | 4 | 6 | 8 |
|---|---|---|---|---|---|---|
| AdamW-Win | 77.9 | 78.0 | 78.0 | 77.9 | 78.1 | 78.0 |
| LAMB-Win | 78.3 | 78.4 | 78.4 | 78.4 | 78.5 | 78.3 |

rithms on ResNet50 by taking AdamW-Win and LAMB-Win as examples because of their superior performance. Table 4 shows the stable performance of AdamW-Win and LAMB-Win when tuning $\gamma$ in a relatively large range, thus validating their robustness to the hyper-parameter $\gamma$.

## 5.2 RESULTS ON NATURAL LANGUAGE MODELING TASKS

**Results on LSTM.** We follow AdaBelief to test our accelerated algorithms via training three-layered LSTM (Schmidhuber et al., 1997) on the Penn TreeBank dataset (Marcinkiewicz, 1994) for 200 epochs. See optimization and training details in Appendix B.

From Table 5, one can observe that our Win-accelerated algorithms consistently surpass the corresponding non-accelerated counterparts, and actually bring 1.2 over-all average perplexity improvement over the four non-accelerated counterparts.

Table 5: Test perplexity of LSTM on Penn Treebank. $*$ is reported by Ad-aBelief (Zhuang et al., 2020).

| AdaBound | 63.6* | Radam | 70.0* |
|---|---|---|---|
| Yogi | 67.5* | AdaBelief | 61.2* |
| fromage | 68.0* | MSVAG | 65.3* |
| SGD-H | 67.4 | Padam | 63.2* |

| SGD-M | 63.8* | Adam | 64.3* |
|---|---|---|---|
| SGD-Win | 61.6$_{+2.2}$ | Adam-Win | 62.7$_{+1.6}$ |

| AdamW | 67.0* | LAMB | 66.8 |
|---|---|---|---|
| AdamW-Win | 66.5$_{+0.5}$ | LAMB-Win | 66.2$_{+0.6}$ |

**Results on Transformer-XL.** We adopt a widely used language sequence model, i.e. Transformer-XL (Dai et al., 2019), to further evaluate the performance of our accelerated algorithms. Since 1) Adam is the most popular and used optimizer in NLP models, including Transformer-XL, and 2) our limited resource cannot well tune the hyper-parameters of other optimizers in Sec. 5.1, we take Adam as an example to show the superiority of our accelerated

Table 6: Test PPL of Transformer-XL-base on WikiText-103 where Adam is the official optimizer. * is reported in the official implementation.

| Transformer-XL | Training Steps | | | |
|---|---|---|---|---|
| | 50k | 100k | 200k | avg. |
| Adam | 28.5 | 25.5 | 24.2* | 26.7 |
| Adam-Win | 26.7 | 25.0 | 24.0 | 25.2$_{+1.5}$ |

algorithms. Follow the official setting of Transformer-XL-base, we use Adam-Win with the default hyper-parameters of Adam on the WikiText-103 dataset. See more details in Appendix B.

Table 6 shows that under different training steps, our accelerated Adam-Win always achieves lower test PPL than the official Adam optimizer. Specifically, it improves 1.5 average test PPL over Adam on the three test cases. All these results are consistent with observations on vision tasks, and they together demonstrate the advantages of our accelerated algorithms.

## 6 CONCLUSION

In this work, we adopt proximal point method to derive a weight-decay-integrated Nesterov acceleration for AdamW and Adam, and extend it to LAMB and SGD. Moreover, we prove the convergence of our accelerated algorithms, i.e. accelerated AdamW, Adam and SGD, and observe the superiority of the accelerated Adam-type algorithm over the vanilla ones in terms of stochastic gradient complexity. Finally, experimental results validate the advantages of our accelerated algorithms.

## ACKNOWLEDGE

Xingyu Xie was supported by National Key R&D Program of China (2022ZD0160302) and the National Natural Science Foundation of China (No. 62276004).

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

APPENDIX

The appendix is structured as follows. In Appendix A, we provide more details of LAMB and Win-accelerated LAMB. Then, Appendix B provides more experimental details, such as hyper-parameter settings of the four accelerated algorithms and the official data augmentations. In Appendix C, we define some necessary notations for our analysis. Then Appendix D provides some auxiliary lemmas throughout this document. Then Appendix E presents the proof of the convergence results in Sec. 1, i.e., the proof of Theorems 1 and 2. Finally, Appendix G provides the proofs of some auxiliary lemmas in Appendix D.

## A    MORE DETAILS OF LAMB AND WIN-ACCELERATED LAMB

Here we introduce more details of vanilla LAMB (You et al., 2019) and our Win-accelerated LAMB. Specifically, Algorithm 2 and 3 respectively summarize the algorithmic steps of LAMB and Win-accelerated LAMB.

## B    MORE EXPERIMENTAL DETAILS

Due to space limitation, we defer the experimental details, such as hyper-parameter settings of the four accelerated algorithms, and their official augmentations in (He et al., 2016) and (Wightman et al., 2021), to this section.

For accelerated algorithms, including AdamW-Win, LAMB-Win, Adam-Win and SGD-Win, always share the default optimizer-inherent hyper-parameters of the vanilla optimizers and its reckless step $\bar{\eta}_k$ is always $2\times$ larger than its conservative step $\eta_k$ for all iterations, i.e. $\bar{\eta}_k = 2\eta_k$. For AdamW-Win, LAMB-Win, Adam-Win, their first- and second-order moment parameters $\beta_1$ and $\beta_2$ are set to the default values $\beta_1 = 0.9$ and $\beta_2 = 0.999$ used in AdamW, LAMB and Adam. For LAMB-Win, its other key parameters, such as "grad averaging" and "trust clip", also adopt the default ones in vanilla LAMB. For SGD-Win, it uses the default momentum parameter 0.9 and set dampening parameter as 0.0 used in vanilla SGD.

---

**Algorithm 2: LAMB in (You et al., 2019)**

**Input:** initialization $\boldsymbol{x}_0 = \boldsymbol{z}_0 = \boldsymbol{0}$, step size $\{(\eta_k, \bar{\eta}_k)\}_{k=0}^T$, moment parameters $\{\beta_1, \beta_2\}$.
**Output:** $\bar{\boldsymbol{x}}$ uniformly seleted from $\{\boldsymbol{x}_k\}_{k=0}^T$.

1  **while** $k < T$ **do**
2  $\quad \boldsymbol{g}_k = \frac{1}{b}\sum_{i=1}^b \nabla f(\boldsymbol{z}_k; \boldsymbol{\zeta}_i)$
3  $\quad \boldsymbol{m}_k = (1 - \beta_1)\boldsymbol{m}_{k-1} + \beta_1 \boldsymbol{g}_k$  /* $\boldsymbol{m}_0 = \boldsymbol{g}_0$ */
4  $\quad \boldsymbol{v}_k = (1 - \beta_2)\boldsymbol{v}_{k-1} + \beta_2 \boldsymbol{g}_k^2$  /* $\boldsymbol{v}_0 = \boldsymbol{g}_0^2$ */
5  $\quad \boldsymbol{u}_k = \frac{\|\boldsymbol{x}_k\|_2}{\|\boldsymbol{m}_k/\sqrt{\boldsymbol{v}_k+\nu}+\lambda\boldsymbol{x}_k\|_2}\left(\frac{\boldsymbol{m}_k}{\sqrt{\boldsymbol{v}_k+\nu}} + \lambda\boldsymbol{x}_k\right)$
6  $\quad \boldsymbol{x}_{k+1} = \boldsymbol{x}_k - \eta_k \boldsymbol{u}_k$
7  **end while**

---

**Algorithm 3: Win-Accelerated LAMB**

**Input:** initialization $\boldsymbol{x}_0 = \boldsymbol{z}_0 = \boldsymbol{0}$, step size $\{(\eta_k, \bar{\eta}_k)\}_{k=0}^T$, moment parameters $\{\beta_1, \beta_2\}$.
**Output:** $(\bar{\boldsymbol{x}}, \bar{\boldsymbol{z}})$ uniformly seleted from $\{(\boldsymbol{x}_k, \boldsymbol{z}_k)\}_{k=0}^T$.

1  **while** $k < T$ **do**
2  $\quad \boldsymbol{g}_k = \frac{1}{b}\sum_{i=1}^b \nabla f(\boldsymbol{z}_k; \boldsymbol{\zeta}_i)$
3  $\quad \boldsymbol{m}_k = (1 - \beta_1)\boldsymbol{m}_{k-1} + \beta_1 \boldsymbol{g}_k$  /* $\boldsymbol{m}_0 = \boldsymbol{g}_0$ */
4  $\quad \boldsymbol{v}_k = (1 - \beta_2)\boldsymbol{v}_{k-1} + \beta_2 \boldsymbol{g}_k^2$  /* $\boldsymbol{v}_0 = \boldsymbol{g}_0^2$ */
5  $\quad \boldsymbol{u}_k = \frac{\|\boldsymbol{x}_k\|_2}{\|\boldsymbol{m}_k/\sqrt{\boldsymbol{v}_k+\nu}+\lambda_k\boldsymbol{x}_k\|_2}\left(\frac{\boldsymbol{m}_k}{\sqrt{\boldsymbol{v}_k+\nu}} + \lambda_k\boldsymbol{x}_k\right)$
6  $\quad \boldsymbol{x}_{k+1} = \frac{1}{1+\lambda_k\eta_k}\left(\boldsymbol{x}_k - \eta_k \boldsymbol{u}_k\right)$ where $\lambda_k = 0$ here
7  $\quad \boldsymbol{z}_{k+1} = \bar{\eta}_k\tau_k \boldsymbol{x}_{k+1} + \eta_k\tau_k\left(\boldsymbol{z}_k - \bar{\eta}_k \boldsymbol{u}_k\right)$ with $\tau_k = \frac{1}{\eta_k+\bar{\eta}_k+\lambda_k\eta_k\bar{\eta}_k}$ and $\lambda_k = 0$ here
8  **end while**

---

**Settings on ResNet18.** Here we follow the conventional supervised training setting used in ResNets (He et al., 2016) and evaluate our accelerated algorithms on ImageNet (Fei-Fei, 2009). For data augmentation in (He et al., 2016), it uses random crop and horizontal flipping with probability 0.5. For warm-up epochs, for all four accelerated algorithms, we set it as 5.0. For base learning rate, we respectively set it as $3 \times 10^{-3}$, $5 \times 10^{-3}$, $3 \times 10^{-3}$, and 1.2 for AdamW-Win, LAMB-Win, Adam-Win and SGD-Win. Moreover, we follow the default setting and use cosine learning rate decay. For weight decay, we respectively set it as $5 \times 10^{-2}$, $5 \times 10^{-2}$, $10^{-6}$, and $10^{-3}$ for AdamW-Win, LAMB-Win, Adam-Win and SGD-Win. On ResNet18, all algorithms are trained for 90 epochs with minibatch size 512 by following the conventional setting.

**Settings on ResNet50&101.** For these two networks, we use "A2 training recipe" in (Wightman et al., 2021) to train them, since this training setting uses stronger data augmentation and largely improves CNNs' performance. Specifically, the data augmentation in (Wightman et al., 2021) uses random crop, horizontal flipping with probability, Mixup with parameter 0.1 (Zhang et al., 2018), CutMix with parameter 1.0 and probability 0.5 (Yun et al., 2019), and RandAugment (Cubuk et al., 2020) with $M = 7, N = 2$ and MSTD $= 0.5$. Moreover, it often use binary cross-entropy (BCE) loss for training.

On both ResNet50 and ResNet101, for base learning rate, we respectively set it as $2 \times 10^{-3}, 8 \times 10^{-3}$, $1 \times 10^{-3}$, and 0.8 for AdamW-Win, LAMB-Win, Adam-Win and SGD-Win. Moreover, we follow the default setting and use cosine learning rate decay. On both ResNet50 and ResNet101, for weight decay, we respectively set it as $5 \times 10^{-2}, 2 \times 10^{-2}, 10^{-5}$, and $5 \times 10^{-4}$ for AdamW-Win, LAMB-Win, Adam-Win and SGD-Win. On both ResNet50 and ResNet101, for warm-up epoch number, we respectively set it as 5, 5, 10, 5 for AdamW-Win, LAMB-Win, Adam-Win and SGD-Win.

**Settings on ViT and PoolFormer.** We follow the widely used official training setting of ViTs (Touvron et al., 2021; Yu et al., 2021). For this setting, data augmentation includes random crop, horizontal flipping with probability, Mixup with parameter 0.8 (Zhang et al., 2018), CutMix with parameter 1.0 and probability 0.5 (Yun et al., 2019), RandAugment (Cubuk et al., 2020) with $M = 9, N = 2$ and MSTD $= 0.5$, and Random Erasing with parameter $p = 0.25$. For training loss, we use cross entropy loss.

On both ViT-S and ViT-B, for base learning rate, we respectively set it as $2 \times 10^{-3}$, $5 \times 10^{-3}$, $1 \times 10^{-4}$, and 0.8 for AdamW-Win, LAMB-Win, Adam-Win and SGD-Win. Moreover, we follow the default setting and use cosine learning rate decay. On both ResNet50 and ResNet101, for weight decay, we respectively set it as $5 \times 10^{-2}, 2 \times 10^{-2}, 10^{-5}$, and $5 \times 10^{-4}$ for AdamW-Win, LAMB-Win, Adam-Win and SGD-Win. On both ResNet50 and ResNet101, for warm-up epoch number, we respectively set it as 5, 60, 30, 5 for AdamW-Win, LAMB-Win, Adam-Win and SGD-Win. For AdamW-Win, following the default setting in AdamW, its minibatch size is 1024 for ViT-S and 512 for ViT-B. For all other accelerated optimizer, their minibatch sizes are always 1024.

**Settings on LSTM.** On LSTM, for base learning rate, we respectively set it as $1 \times 10^{-3}$, $1 \times 10^{-2}$, $1 \times 10^{-2}$, and 15.0 for AdamW-Win, LAMB-Win, Adam-Win and SGD-Win. Moreover, we follow the default setting and divide the learning rate by 10 at epoch 100 and 145. For weight decay, we respectively set it as $2 \times 10^{-2}, 5 \times 10^{-2}, 1.8 \times 10^{-6}$, and $2 \times 10^{-5}$ for AdamW-Win, LAMB-Win, Adam-Win and SGD-Win. We do not utilize the warmup strategy in this experiment. Following the default setting, we set minibatch size as 20.

**Settings on Transformer-XL.** On Transformer-XL, for base learning rate, we set it as $4 \times 10^{-4}$ for Adam-Win. Moreover, we follow the default setting and use cosine learning rate decay. For weight decay, we set it as $10^{-6}$ for Adam-Win. For warm-up steps, we set it as 2000. Following the default setting, we set minibatch size as $60 \times 4$.

**Test accuracy curves of SGD-Win and Adam-Win on ResNet18.** Here we investigate the convergence behaviors of our accelerated algorithms and hope to explain their better test performance over their non-accelerated counterparts. In each sub-figure pair of Fig. 1, we plot the curves of training and test losses along with the training epochs on ResNet18 and ViT-B. One can find that our accelerated algorithms, *e.g.* AdamW-Win, show much faster convergence behaviors than their non-accelerated counterparts, *e.g.* AdamW. Moreover, SGD-Win also converges faster than Nesterove-accelerated SGD, i.e. SGD-M. We also plot the curves of test accuracy in Fig. 2, showing the superior convergence speed of AdamW-Win and LAMB-Win over their corresponding non-accelerated versions. Fig. 3 in Appendix B also reveals SGD-Win and Adam-Win enjoy faster convergence than

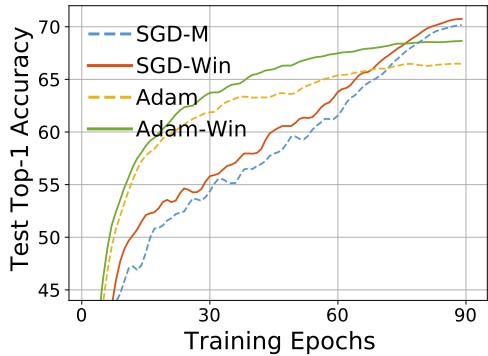

Figure 3: Test accuracy curve of SGD-Win and Adam-Win on ResNet18. See the curves of AdamW-Win and LAMB-Win in manuscript.

their non-accelerated counterparts in terms of test accuracy. So these faster convergence behaviors could contribute to our accelerated algorithms for their higher performance over non-accelerated algorithms under the same computational cost.

## C   NOTATIONS

Here we first give some important notations used in this document. For brevity, we let

$$s_k = \sqrt{v_k + \nu}.$$

Since we have $\|m_k\|_\infty \le c_\infty$ and $\nu \le \|v_i + \nu\|_\infty \le c_\infty^2 + \nu$ in Lemma 3 (see Appendix D), for brevity, let

$$c_1 := \nu^{0.5} \le \|s_k\|_\infty \le c_2 := (c_\infty^2 + \nu)^{0.5}.$$

Also we define

$$w_k := m_k + \lambda_k x_k * s_k, \qquad x_{k+1} - x_k = -\frac{\eta_k}{1 + \lambda_k \eta_k} \frac{m_k + \lambda_k x_k * s_k}{s_k} = -\frac{\eta_k}{1 + \lambda_k \eta_k} \frac{w_k}{s_k}.$$

Next, we introduce an virtual sequence $\{y_k\}$ into the algorithm. In this way, we can rewrite the update steps in Algorithm 1 in the manuscript as its equivalent form (12):

$$\begin{cases} g_k = \frac{1}{b} \sum_{i=1}^b \nabla f(z_k; \zeta_i); \\ m_k = (1 - \beta_1) m_{k-1} + \beta_1 g_k; \\ v_k = (1 - \beta_2) v_{k-1} + \beta_2 g_k^2; \\ x_{k+1} = \frac{1}{1 + \lambda_k \eta_k} \left( x_k - \eta_k \frac{m_k}{s_k} \right) \\ y_{k+1} = z_k - \bar{\eta}_k \frac{m_k}{s_k} \\ z_{k+1} = \bar{\eta}_k \tau_k x_{k+1} + \eta_k \tau_k y_{k+1} \end{cases} \tag{12}$$

where $m_0 = g_0$ and $v_0 = g_0^2$.

For analysis, we further define

$$F_k(\theta_k) = F(\theta) + \frac{\lambda_k}{2} \|\theta\|_{s_k}^2 = \mathbb{E}_\zeta[f(\theta; \zeta)] + \frac{\lambda_k}{2} \|\theta\|_{s_k}^2, \tag{13}$$

where $\lambda_k = \lambda(1 - \mu)^k$ in which $\mu = \frac{\beta_2 c_\infty^2}{\delta}$. In the following, we mainly use these notations to finish our proofs.

## D   AUXILIARY LEMMAS

Before giving our analysis, we first provide some important lemmas.

**Lemma 3.** *Suppose the sequence $\{\boldsymbol{x}_k, \boldsymbol{y}_k, \boldsymbol{z}_k\}$ are updated by Eqn. (12). That is, $\boldsymbol{x}_{k+1} = \frac{1}{1+\lambda_k\eta_k}\left(\boldsymbol{x}_k - \eta_k\frac{\boldsymbol{m}_k}{\boldsymbol{s}_k}\right), \boldsymbol{y}_{k+1} = \boldsymbol{z}_k - \bar{\eta}_k\frac{\boldsymbol{m}_k}{\boldsymbol{s}_k}, \boldsymbol{z}_{k+1} = \bar{\eta}_k\tau_k\boldsymbol{x}_{k+1} + \eta_k\tau_k\boldsymbol{y}_{k+1}, \boldsymbol{s}_k = \sqrt{\boldsymbol{v}_k + \nu}.$ Then $\{(\boldsymbol{m}_k, \boldsymbol{s}_k)\}$ satisfies Assume $c_{s,\infty} \leq \|g_k\|_\infty \leq c_\infty$, then we have*

$$\|\boldsymbol{m}_k\|_\infty \leq c_\infty, \quad \|\boldsymbol{v}_i + \nu\|_\infty \leq c_\infty^2 + \nu, \quad \frac{\beta_2 c_\infty^2}{2(c_{s,\infty}^2 + \nu)} \leq \left\|\frac{\boldsymbol{s}_k}{\boldsymbol{s}_{k+1}}\right\|_\infty < 1 + \frac{\beta_2 c_\infty^2}{2(c_{s,\infty}^2 + \nu)}.$$

See its proof in Appendix G.1.

**Lemma 4.** *(Xie et al., 2022) Suppose the sequence $\{\boldsymbol{x}_k, \boldsymbol{y}_k, \boldsymbol{z}_k\}$ are updated by Eqn. (12). That is, $\boldsymbol{x}_{k+1} = \frac{1}{1+\lambda_k\eta_k}\left(\boldsymbol{x}_k - \eta_k\frac{\boldsymbol{m}_k}{\boldsymbol{s}_k}\right), \boldsymbol{y}_{k+1} = \boldsymbol{z}_k - \bar{\eta}_k\frac{\boldsymbol{m}_k}{\boldsymbol{s}_k}, \boldsymbol{z}_{k+1} = \bar{\eta}_k\tau_k\boldsymbol{x}_{k+1} + \eta_k\tau_k\boldsymbol{y}_{k+1}, \boldsymbol{s}_k = \sqrt{\boldsymbol{v}_k + \nu}.$ Then $\{\boldsymbol{z}_k\}$ satisfies*

$$\mathbb{E}\left[\|\boldsymbol{m}_k - \nabla F(\boldsymbol{z}_k)\|^2\right]$$
$$\leq (1 - \beta_1)\mathbb{E}\left[\|\boldsymbol{m}_{k-1} - \nabla F(\boldsymbol{z}_{k-1})\|^2\right] + \frac{(1-\beta_1)^2 L^2}{\beta_1}\mathbb{E}\left[\|\boldsymbol{z}_k - \boldsymbol{z}_{k-1}\|^2\right] + \frac{\beta_1^2\sigma^2}{b}.$$

**Lemma 5.** *Suppose the sequence $\{\boldsymbol{x}_k, \boldsymbol{y}_k, \boldsymbol{z}_k\}$ are updated by Eqn. (12). That is, $\boldsymbol{x}_{k+1} = \frac{1}{1+\lambda_k\eta_k}\left(\boldsymbol{x}_k - \eta_k\frac{\boldsymbol{m}_k}{\boldsymbol{s}_k}\right), \boldsymbol{y}_{k+1} = \boldsymbol{z}_k - \bar{\eta}_k\frac{\boldsymbol{m}_k}{\boldsymbol{s}_k}, \boldsymbol{z}_{k+1} = \bar{\eta}_k\tau_k\boldsymbol{x}_{k+1} + \eta_k\tau_k\boldsymbol{y}_{k+1}, \boldsymbol{s}_k = \sqrt{\boldsymbol{v}_k + \nu}.$ By setting $\eta_k = \eta, \bar{\eta}_k = \bar{\eta}, \beta_{1,k} = \beta_1$ and $\beta_{2,k} = \beta_2$, then we have*

$$\boldsymbol{y}_{k+1} - (1 + \lambda_k\bar{\eta})\boldsymbol{x}_{k+1} = -\rho_{k+1}\sum_{i=0}^{k}\frac{1}{\rho_{i+1}}\frac{\bar{\eta} - \eta}{1 + \lambda_i\eta}\frac{\boldsymbol{w}_i}{\boldsymbol{s}_i},$$

$$\|\boldsymbol{y}_{k+1} - (1 + \lambda_k\bar{\eta})\boldsymbol{x}_{k+1}\|^2 \leq \rho_{k+1}(\bar{\eta} - \eta)^2\sum_{i=0}^{k}\frac{1}{\rho_{i+1}(1 - \eta\tau_{i-1})(1 + \lambda_i\eta)^2}\left\|\frac{\boldsymbol{w}_i}{\boldsymbol{s}_i}\right\|^2,$$

$$\|\boldsymbol{z}_{k+1} - \boldsymbol{x}_{k+1}\|^2 \leq \tau_k\rho_{k+1}\eta(\bar{\eta} - \eta)^2\sum_{i=0}^{k}\frac{1}{\rho_{i+1}(1 - \eta\tau_{i-1})(1 + \lambda_i\eta)^2}\left\|\frac{\boldsymbol{w}_i}{\boldsymbol{s}_i}\right\|^2,$$

$$\|\boldsymbol{z}_{k+1} - \boldsymbol{z}_k\|^2 \leq \frac{2\bar{\eta}^2}{(1 + \lambda_k\eta)^2}\left\|\frac{\boldsymbol{w}_k}{\boldsymbol{s}_k}\right\|^2$$
$$+ 2\rho_{k+1}\bar{\eta}^2(\bar{\eta} - \eta)^2\tau_k^2(1 + \lambda_k\eta)^2\sum_{i=0}^{k}\frac{1}{\rho_{i+1}(1 - \eta\tau_{i-1})(1 + \lambda_i\eta)^2}\left\|\frac{\boldsymbol{w}_i}{\boldsymbol{s}_i}\right\|^2,$$

*where $\rho_{k+1} = \eta\tau_{k-1}\rho_k$, $\rho_1 = 1$ and $\rho_0 = 0$.*

See its proof in Appendix G.2.

**Lemma 6.** *Suppose the sequence $\{\boldsymbol{x}_k, \boldsymbol{y}_k, \boldsymbol{z}_k\}$ are updated by Eqn. (12). That is, $\boldsymbol{x}_{k+1} = \frac{1}{1+\lambda\eta_k}\left(\boldsymbol{x}_k - \eta_k\frac{\boldsymbol{m}_k}{\boldsymbol{s}_k}\right), \boldsymbol{y}_{k+1} = \boldsymbol{z}_k - \bar{\eta}_k\frac{\boldsymbol{m}_k}{\boldsymbol{s}_k}, \boldsymbol{z}_{k+1} = \bar{\eta}_k\tau_k\boldsymbol{x}_{k+1} + \eta_k\tau_k\boldsymbol{y}_{k+1}, \boldsymbol{s}_k = \sqrt{\boldsymbol{v}_k + \nu}.$ By setting $\eta_k = \eta, \bar{\eta}_k = \bar{\eta}, \beta_{1,k} = \beta_1$ and $\beta_{2,k} = \beta_2$, then we have*

$$\mathbb{E}\left[\|\boldsymbol{m}_k - \nabla F(\boldsymbol{x}_k)\|^2\right] \leq 2(1 - \beta_1)\mathbb{E}\left[\|\boldsymbol{m}_{k-1} - \nabla F(\boldsymbol{z}_{k-1})\|^2\right] + \frac{2\Pi_k(1-\beta_1)^2 L^2}{\beta_1} + \frac{2\beta_1^2\sigma^2}{b} + 2L\Pi_k',$$

*where*

$$\Pi_k := \frac{2\bar{\eta}^2}{(1 + \lambda_{k-1}\eta)^2}\left\|\frac{\boldsymbol{w}_{k-1}}{\boldsymbol{s}_{k-1}}\right\|^2 + 2\rho_k\bar{\eta}^2(\bar{\eta} - \eta)^2\tau_{k-1}^2(1 + \lambda_{k-1}\eta)^2\sum_{i=0}^{k-1}\frac{1}{\rho_{i+1}(1 - \eta\tau_{i-1})(1 + \lambda_i\eta)^2}\left\|\frac{\boldsymbol{w}_i}{\boldsymbol{s}_i}\right\|^2,$$

$$\Pi_k' := \tau_{k-1}\rho_k\eta(\bar{\eta} - \eta)^2\sum_{i=0}^{k-1}\frac{1}{\rho_{i+1}(1 - \eta\tau_{i-1})(1 + \lambda_i\eta)^2}\left\|\frac{\boldsymbol{w}_i}{\boldsymbol{s}_i}\right\|^2,$$

*where $\rho_{k+1} = \eta\tau_{k-1}\rho_k$, $\rho_1 = 1$ and $\rho_0 = 0$.*

see its proof in Appendix G.3.

# E  PROOF OF THEOREM 1

*Proof.* Recall our definition $F_k(z_k) = F(z) + \frac{\lambda_k}{2}\|z\|_{s_k}^2 = \mathbb{E}_\zeta[f(z;\zeta)] + \frac{\lambda_k}{2}\|z\|_{s_k}^2$, in the (13). By using the smoothness of $f(\theta;\zeta)$, we can obtain

$$F_{k+1}(x_{k+1})$$

$$\leq F(x_k) + \langle \nabla F(x_k), x_{k+1} - x_k\rangle + \frac{L}{2}\|x_{k+1} - x_k\|^2 + \frac{\lambda_{k+1}}{2}\|x_{k+1}\|_{s_{k+1}}^2$$

$$\overset{①}{\leq} F(x_k) + \langle \nabla F(x_k), x_{k+1} - x_k\rangle + \frac{L}{2}\|x_{k+1} - x_k\|^2 + \frac{\lambda_{k+1}}{2(1-\mu)}\|x_{k+1}\|_{s_k}^2$$

$$\overset{②}{\leq} F(x_k) + \frac{\lambda_k}{2}\|x_k\|_{s_k}^2 + \langle \nabla F(x_k) + \lambda_k x_k * s_k, x_{k+1} - x_k\rangle + \frac{L}{2}\|x_{k+1} - x_k\|^2 + \frac{\lambda_k}{2}\|x_{k+1} - x_k\|_{s_k}^2$$

$$= F_k(x_k) - \frac{\eta_k}{1+\lambda_k\eta_k}\left\langle \nabla F(x_k) + \lambda_k x_k * s_k, \frac{w_k}{s_k}\right\rangle + \frac{L\eta_k^2}{2(1+\lambda_k\eta_k)^2}\left\|\frac{w_k}{s_k}\right\|^2 + \frac{\lambda_k\eta_k^2}{2(1+\lambda_k\eta_k)^2}\left\|\frac{w_k}{s_k}\right\|_{s_k}^2$$

$$= F_k(x_k) + \frac{1}{2}\left\|\sqrt{\frac{\eta_k}{(1+\lambda_k\eta_k)s_k}}\left(\nabla F(x_k) + \lambda_k x_k * s_k - w_k\right)\right\|^2 - \frac{1}{2}\left\|\sqrt{\frac{\eta_k}{(1+\lambda_k\eta_k)s_k}}\left(\nabla F(x_k) + \lambda_k x_k * s_k\right)\right\|^2$$

$$\quad - \frac{1}{2}\left\|\sqrt{\frac{\eta_k}{(1+\lambda_k\eta_k)s_k}}w_k\right\|^2 + \frac{L\eta_k^2}{2(1+\lambda_k\eta_k)^2}\left\|\frac{w_k}{s_k}\right\|^2 + \frac{\lambda_k\eta_k^2}{2(1+\lambda_k\eta_k)^2}\left\|\frac{w_k}{s_k}\right\|_{s_k}^2$$

$$\overset{③}{\leq} F_k(x_k) + \frac{\eta_k}{2c_1(1+\lambda_k\eta_k)}\|\nabla F(x_k) - m_k\|^2 - \frac{\eta_k}{2c_2(1+\lambda_k\eta_k)}\|\nabla F_k(x_k)\|^2$$

$$\quad - \frac{\eta_k}{2c_2(1+\lambda_k\eta_k)}\left[1 - \frac{c_2L\eta_k}{c_1^2(1+\lambda_k\eta_k)} - \frac{c_2\lambda_k\eta_k}{c_1(1+\lambda_k\eta_k)}\right]\|w_k\|^2$$

$$\overset{④}{\leq} F_k(x_k) + \frac{\eta_k}{2c_1(1+\lambda_k\eta_k)}\|\nabla F(x_k) - m_k\|^2 - \frac{\eta_k}{2c_2(1+\lambda_k\eta_k)}\|\nabla F_k(x_k)\|^2 - \frac{\eta_k}{4c_2(1+\lambda_k\eta_k)}\|w_k\|^2,$$

where ① holds since Lemma 3 proves $\left\|\frac{s_k}{s_{k+1}}\right\|_\infty \in [1-\mu, 1+\mu]$ ($\forall p \in [0,1]$) in which $\mu = \frac{\beta_2 c_\infty^2}{\nu}$; ② holds because $\lambda_k = \frac{\lambda_{k+1}}{1-\mu}$ and

$$\|x_{k+1}\|_{s_k}^2 = \|x_k\|_{s_k}^2 + \|x_{k+1} - x_k\|_{s_k}^2 + 2\langle x_{k+1} - x_k, x_k\rangle_{s_k}.$$

③ holds, because

$$w_k := m_k + \lambda_k x_k * s_k, \qquad x_{k+1} - x_k = -\frac{\eta_k}{1+\lambda_k\eta_k}\frac{m_k + \lambda_k x_k * s_k}{s_k} = -\frac{\eta_k}{1+\lambda_k\eta_k}\frac{w_k}{s_k},$$

$$c_1 := \nu^{0.5} \leq \|s_k\|_\infty \leq c_2 := (c_\infty^2 + \nu)^{0.5}.$$

④ holds, since we set $\eta_k \leq \frac{c_1^2(1+\lambda_k\eta_k)}{2c_2(L+\lambda_k c_1)}$ such that $\frac{c_2L\eta_k}{c_1^2(1+\lambda_k\eta_k)} + \frac{c_2\lambda_k\eta_k}{c_1(1+\lambda_k\eta_k)} \leq \frac{1}{2}$.

From Lemma 6, by setting $\eta_k = \eta$, $\bar\eta_k = \bar\eta$ and $\beta_{1,k} = \beta_1$, we have

$$\mathbb{E}\left[\|m_k - \nabla F(x_k)\|^2\right] \leq 2(1-\beta_1)\mathbb{E}\left[\|m_{k-1} - \nabla F(z_{k-1})\|^2\right] + \frac{2\Pi_k(1-\beta_1)^2L^2}{\beta_1} + \frac{2\beta_1^2\sigma^2}{b} + 2L\Pi_k', \tag{14}$$

where

$$\Pi_k := \frac{2\bar\eta^2}{(1+\lambda_{k-1}\eta)^2}\left\|\frac{w_{k-1}}{s_{k-1}}\right\|^2 + 2\rho_k\bar\eta^2(\bar\eta-\eta)^2\tau_{k-1}^2(1+\lambda_{k-1}\eta)^2\sum_{i=0}^{k-1}\frac{1}{\rho_{i+1}(1-\eta\tau_{i-1})(1+\lambda_i\eta)^2}\left\|\frac{w_i}{s_i}\right\|^2,$$

$$\Pi_k' := \tau_{k-1}\rho_k\eta(\bar\eta-\eta)^2\sum_{i=0}^{k-1}\frac{1}{\rho_{i+1}(1-\eta\tau_{i-1})(1+\lambda_i\eta)^2}\left\|\frac{w_i}{s_i}\right\|^2,$$

$$\tag{15}$$

Here $\rho_{k+1} = \eta\tau_{k-1}\rho_k$, $\rho_1 = 1$ and $\rho_0 = 0$. By considering $c_2 \geq \|\boldsymbol{s}_k\|_\infty \geq c_1$, we have

$$
\Pi_k \leq \bar{\Pi}_k := \frac{2\bar{\eta}^2}{c_1^2(1+\lambda_{k-1}\eta)^2} \|\boldsymbol{w}_{k-1}\|^2 + \frac{2\rho_k\bar{\eta}^2(\bar{\eta}-\eta)^2\tau_{k-1}^2(1+\lambda_{k-1}\eta)^2}{c_1^2} \sum_{i=0}^{k-1} \frac{1}{\rho_{i+1}(1-\eta\tau_{i-1})(1+\lambda_i\eta)^2} \|\boldsymbol{w}_i\|^2,
$$

$$
\Pi_k' \leq \bar{\Pi}_k' := \frac{\tau_{k-1}\rho_k\eta(\bar{\eta}-\eta)^2}{c_1^2} \sum_{i=0}^{k-1} \frac{1}{\rho_{i+1}(1-\eta\tau_{i-1})(1+\lambda_i\eta)^2} \|\boldsymbol{w}_i\|^2,
$$

$$(16)$$

Therefore, by plugging the results in Eqn. (14) into the upper bound of $F_{k+1}(\boldsymbol{x}_{k+1})$, we have

$$
\begin{aligned}
&F_{k+1}(\boldsymbol{x}_{k+1}) \\
\leq& F_k(\boldsymbol{x}_k) - \frac{\eta}{2c_2(1+\lambda_k\eta)} \|\nabla F_k(\boldsymbol{x}_k)\|^2 - \frac{\eta}{4c_2(1+\lambda_k\eta)} \|\boldsymbol{w}_k\|^2 \\
&+ \frac{\eta(1-\beta_1)}{c_1(1+\lambda_k\eta)} \mathbb{E}\left[\|\boldsymbol{m}_{k-1} - \nabla F(\boldsymbol{z}_{k-1})\|^2\right] + \frac{\eta\bar{\Pi}_k(1-\beta_1)^2L^2}{c_1\beta_1(1+\lambda_k\eta)} + \frac{\eta\beta_1^2\sigma^2}{c_1(1+\lambda_k\eta)b} + \frac{\eta L\bar{\Pi}_k'}{c_1(1+\lambda_k\eta)} \\
\overset{①}{\leq}& F_k(\boldsymbol{x}_k) - \frac{\eta}{2c_2(1+\lambda_k\eta)} \|\nabla F_k(\boldsymbol{x}_k)\|^2 - \frac{\eta}{4c_2(1+\lambda_k\eta)} \|\boldsymbol{w}_k\|^2 \\
&+ \frac{\eta(1-\beta_1)}{c_1} \mathbb{E}\left[\|\boldsymbol{m}_{k-1} - \nabla F(\boldsymbol{z}_{k-1})\|^2\right] + \frac{\eta\bar{\Pi}_k(1-\beta_1)^2L^2}{c_1\beta_1(1+\lambda_k\eta)} + \frac{\eta\beta_1^2\sigma^2}{c_1(1+\lambda_k\eta)b} + \frac{\eta L\bar{\Pi}_k'}{c_1(1+\lambda_k\eta)},
\end{aligned}
$$

$$(17)$$

where ① uses the fact that $0 < \lambda_k \leq \lambda$. Then, from Lemma 4, we have

$$
\begin{aligned}
&\mathbb{E}\left[\|\boldsymbol{m}_k - \nabla F(\boldsymbol{z}_k)\|^2\right] \\
\leq& (1-\beta_1)\mathbb{E}\left[\|\boldsymbol{m}_{k-1} - \nabla F(\boldsymbol{z}_{k-1})\|^2\right] + \frac{(1-\beta_1)^2L^2}{\beta_1}\mathbb{E}\left[\|\boldsymbol{z}_k - \boldsymbol{z}_{k-1}\|^2\right] + \frac{\beta_1^2\sigma^2}{b} \\
\overset{①}{\leq}& (1-\beta_1)\mathbb{E}\left[\|\boldsymbol{m}_{k-1} - \nabla F(\boldsymbol{z}_{k-1})\|^2\right] + \frac{(1-\beta_1)^2L^2\bar{\Pi}_k}{\beta_1} + \frac{\beta_1^2\sigma^2}{b}
\end{aligned}
$$

$$(18)$$

where we use the results in Lemma 5 that

$$
\|\boldsymbol{z}_k - \boldsymbol{z}_{k-1}\|^2 \leq \Pi_k \leq \bar{\Pi}_k.
$$

Then we add Eqn. (17) and $\alpha\times$ (18) as follows:

$$
\begin{aligned}
&F_{k+1}(\boldsymbol{x}_{k+1}) + \alpha\mathbb{E}\left[\|\boldsymbol{m}_k - \nabla F(\boldsymbol{z}_k)\|^2\right] \\
\leq& F_k(\boldsymbol{x}_k) - \frac{\eta}{2c_2(1+\lambda_k\eta)} \|\nabla F_k(\boldsymbol{x}_k)\|^2 - \frac{\eta}{4c_2(1+\lambda_k\eta)} \|\boldsymbol{w}_k\|^2 \\
&+ (1-\beta_1)\left(\frac{\eta}{c_1} + \alpha\right) \mathbb{E}\left[\|\boldsymbol{m}_{k-1} - \nabla F(\boldsymbol{z}_{k-1})\|^2\right] + \frac{\eta\bar{\Pi}_k(1-\beta_1)^2L^2}{c_1\beta_1(1+\lambda_k\eta)} + \frac{\eta\beta_1^2\sigma^2}{c_1(1+\lambda_k\eta)b} \\
&+ \frac{\eta L\bar{\Pi}_k'}{c_1(1+\lambda_k\eta)} + \frac{\alpha(1-\beta_1)^2L^2\bar{\Pi}_k}{\beta_1} + \frac{\alpha\beta_1^2\sigma^2}{b}
\end{aligned}
$$

$$(19)$$

Then by setting $\alpha = \frac{\eta(1-\beta_1)}{c_1\beta_1}$ and $G_{k+1}(\boldsymbol{x}_{k+1}) = F_{k+1}(\boldsymbol{x}_{k+1}) + \frac{\eta(1-\beta_1)}{c_1\beta_1}\mathbb{E}\left[\|\boldsymbol{m}_k - \nabla F(\boldsymbol{x}_k)\|^2\right] = \mathbb{E}_{\boldsymbol{\zeta}}[f(\boldsymbol{z};\boldsymbol{\zeta})] + \frac{\lambda_k}{2}\|\boldsymbol{z}\|_{\boldsymbol{s}_k}^2 + \frac{\eta(1-\beta_1)}{c_1\beta_1}\mathbb{E}\left[\|\boldsymbol{m}_k - \nabla F(\boldsymbol{x}_k)\|^2\right]$, we can obtain

$$
\begin{aligned}
G_{k+1}(\boldsymbol{x}_{k+1}) \leq & G_k(\boldsymbol{x}_k) - \frac{\eta}{2c_2(1+\lambda_k\eta)}\|\nabla F_k(\boldsymbol{x}_k)\|^2 - \frac{\eta}{4c_2(1+\lambda_k\eta)}\|\boldsymbol{w}_k\|^2 \\
& + \frac{\eta\bar{\Pi}_k(1-\beta_1)^2 L^2}{c_1\beta_1(1+\lambda_k\eta)} + \frac{\eta\beta_1^2\sigma^2}{c_1(1+\lambda_k\eta)b} + \frac{\eta L\bar{\Pi}_k'}{c_1(1+\lambda_k\eta)} \\
& + \frac{\eta(1-\beta_1)^3 L^2\bar{\Pi}_k}{c_1\beta_1^2} + \frac{\eta(1-\beta_1)\beta_1\sigma^2}{c_1 b} \\
\overset{①}{\leq} & G_k(\boldsymbol{x}_k) - \frac{\eta}{2c_2(1+\lambda_k\eta)}\|\nabla F_k(\boldsymbol{x}_k)\|^2 - \frac{\eta}{4c_2(1+\lambda_k\eta)}\|\boldsymbol{w}_k\|^2 \\
& + \frac{\eta(1-\beta_1)^2 L^2\bar{\Pi}_k}{c_1\beta_1^2} + \frac{\eta L\bar{\Pi}_k'}{c_1(1+\lambda_k\eta)} + \frac{\eta\beta_1\sigma^2}{c_1 b},
\end{aligned}
$$

where ① uses the fact that $0 < \lambda_k \leq \lambda$. Then summing the above inequality from $k = 0$ to $k = T-1$ and using $0 < \lambda_k \leq \lambda$ give

$$
\begin{aligned}
& \frac{1}{T}\sum_{k=0}^{T-1}\mathbb{E}\left[\|\nabla F_k(\boldsymbol{x}_k)\|^2 + \frac{1}{2}\|\boldsymbol{w}_k\|^2\right] \\
\leq & \frac{2c_2(1+\lambda\eta)}{\eta T}[G(\boldsymbol{x}_0) - G(\boldsymbol{x}_T)] + \frac{2c_2\beta_1\sigma^2(1+\lambda\eta)}{c_1 bT} + \frac{2c_2\beta_1^2\sigma^2}{c_1 b} \\
& + \frac{2c_2(1-\beta_1)^2 L^2(1+\lambda\eta)}{c_1\beta_1^2 T}\sum_{k=0}^{T-1}\bar{\Pi}_k + \frac{2c_2 L}{c_1 T}\sum_{k=0}^{T-1}\bar{\Pi}_k' \\
\leq & \frac{2c_2(1+\lambda\eta)\Delta}{\eta T} + \frac{2c_2\beta_1\sigma^2(1+\lambda\eta)}{c_1 b} + \frac{2c_2(1-\beta_1)^2 L^2(1+\lambda\eta)}{c_1\beta_1^2 T}\sum_{k=0}^{T-1}\bar{\Pi}_k + \frac{2c_2 L}{c_1 T}\sum_{k=0}^{T-1}\bar{\Pi}_k'
\end{aligned}
$$

where

$$
\begin{aligned}
& G(\boldsymbol{x}_0) - G(\boldsymbol{x}_T) \\
= & F_0(\boldsymbol{x}_0) + \frac{\eta(1-\beta_1)}{c_1\beta_1}\mathbb{E}\left[\|\boldsymbol{m}_{-1} - \nabla F(\boldsymbol{x}_{-1})\|^2\right] - F_T(\boldsymbol{x}_T) - \frac{\eta(1-\beta_1)}{c_1\beta_1}\mathbb{E}\left[\|\boldsymbol{m}_{T-1} - \nabla F(\boldsymbol{x}_{T-1})\|^2\right] \\
= & F(\boldsymbol{x}_0) + \lambda_0\|\boldsymbol{x}_0\|_{\boldsymbol{s}_0} - F(\boldsymbol{x}_T) - \lambda_T\|\boldsymbol{x}_T\|_{\boldsymbol{s}_T} - \frac{\eta(1-\beta_1)}{c_1\beta_1}\mathbb{E}\left[\|\boldsymbol{m}_{T-1} - \nabla F(\boldsymbol{x}_{T-1})\|^2\right] \\
\leq & F(\boldsymbol{x}_0) - F(\boldsymbol{x}_T) \\
\leq & \Delta
\end{aligned}
$$

where $\Delta = F(\boldsymbol{x}_0) - F(\boldsymbol{x}_\star)$; $\boldsymbol{x}_{-1}$ and $\boldsymbol{m}_{-1}$ are two virtual points which satisfy $\boldsymbol{m}_{-1} = \nabla F(\boldsymbol{x}_{-1})$.
Now we try to bound $\sum_{k=0}^{T-1} \bar{\Pi}_k$ and $\sum_{k=0}^{T-1} \bar{\Pi}_k'$. Firstly, we have

$$
\begin{aligned}
\sum_{k=0}^{T-1} \bar{\Pi}_k &= \sum_{k=0}^{T-1} \left[ \frac{2\bar{\eta}^2}{c_1^2(1+\lambda_{k-1}\eta)^2} \|\boldsymbol{w}_{k-1}\|^2 + \frac{2\rho_k \bar{\eta}^2 (\bar{\eta}-\eta)^2 \tau_{k-1}^2 (1+\lambda_{k-1}\eta)^2}{c_1^2} \sum_{i=0}^{k-1} \frac{1}{\rho_{i+1}(1-\eta\tau_{i-1})(1+\lambda_i\eta)^2} \|\boldsymbol{w}_i\|^2 \right] \\
&\overset{①}{\leq} \frac{2\bar{\eta}^2}{c_1^2} \sum_{k=0}^{T-1} \left[ \|\boldsymbol{w}_{k-1}\|^2 \right] + \frac{2\bar{\eta}^2(\bar{\eta}-\eta)^2}{c_1^2} \sum_{k=0}^{T-1} \rho_k \tau_{k-1}^2 (1+\lambda_{k-1}\eta)^2 \left[ \sum_{i=0}^{k-1} \frac{1}{\rho_{i+1}(1-\eta\tau_{i-1})(1+\lambda_i\eta)^2} \|\boldsymbol{w}_i\|^2 \right] \\
&= \frac{2\bar{\eta}^2}{c_1^2} \sum_{k=0}^{T-1} \left[ \|\boldsymbol{w}_{k-1}\|^2 \right] + \frac{2\bar{\eta}^2(\bar{\eta}-\eta)^2}{c_1^2} \sum_{k=0}^{T-1} \frac{1}{\rho_{k+1}(1-\eta\tau_{k-1})(1+\lambda_k\eta)^2} \|\boldsymbol{w}_k\|^2 \left[ \sum_{i=k}^{T-1} \rho_i \tau_{i-1}^2 (1+\lambda_{i-1}\eta)^2 \right] \\
&\overset{②}{\leq} \frac{2\bar{\eta}^2}{c_1^2} \sum_{k=0}^{T-1} \left[ \|\boldsymbol{w}_{k-1}\|^2 \right] + \frac{2a^2\bar{\eta}^2(\bar{\eta}-\eta)^2}{c_1^2(1-\eta\tau)} \sum_{k=0}^{T-1} \frac{1}{\rho_{k+1}} \|\boldsymbol{w}_k\|^2 \left[ \sum_{i=k}^{T-1} \rho_i \tau_{i-1}^2 \right] \\
&= \frac{2\bar{\eta}^2}{c_1^2} \sum_{k=0}^{T-1} \left[ \|\boldsymbol{w}_{k-1}\|^2 \right] + \frac{2a^2\bar{\eta}^2(\bar{\eta}-\eta)^2\tau}{c_1^2\eta(1-\eta\tau)^2} \sum_{k=0}^{T-1} \left[ \|\boldsymbol{w}_k\|^2 \right] \\
&\overset{③}{\leq} \frac{2\bar{\eta}^2}{c_1^2} \sum_{k=0}^{T-1} \left[ \|\boldsymbol{w}_{k-1}\|^2 \right] + \frac{2a^2\bar{\eta}^2(\bar{\eta}-\eta)^2\tau}{c_1^2\eta(1-\eta\tau)^2} \sum_{k=0}^{T-1} \left[ \|\boldsymbol{w}_k\|^2 \right] \\
&\overset{④}{\leq} \frac{2\gamma^2\eta^2}{c_1^2} \left[ 1 + a^2(1+\gamma)(\gamma-1)^2 \right] \sum_{k=0}^{T-1} \left[ \|\boldsymbol{w}_{k-1}\|^2 \right] \\
&\leq \frac{2\gamma^2\eta^2}{c_1^2} \left[ 1 + a^2\gamma^3 \right] \sum_{k=0}^{T-1} \left[ \|\boldsymbol{w}_{k-1}\|^2 \right],
\end{aligned}
$$

where ① holds since $0 \leq \lambda_k \leq \lambda$; ② holds, since 1) for $i \geq k$ we have $\frac{1+\lambda_{i-1}\eta}{1+\lambda_k\eta} \leq \frac{1+\lambda_{k-1}\eta}{1+\lambda_k\eta} = \frac{1+\lambda_{k-1}\eta}{1+(1-\mu)\lambda_{k-1}\eta} \leq \frac{1+\lambda\eta}{1+(1-\mu)\lambda\eta} = a \leq \frac{1}{1-\mu}$ and 2) $\frac{1}{1-\eta\tau_{i-1}} = \frac{\eta+\bar{\eta}+\lambda_{i-1}\bar{\eta}\eta}{\bar{\eta}+\lambda_{i-1}\bar{\eta}\eta} = 1 + \frac{\eta}{\bar{\eta}+\lambda_{i-1}\bar{\eta}\eta} \leq 1 + \frac{\eta}{\bar{\eta}} = \frac{1}{1-\eta\tau}$ whose minimum is at $\lambda_{i-1} = 0$ and $\tau = \frac{1}{\eta+\bar{\eta}}$; ③ holds, since $\sum_{i=k}^{T-1} \rho_i \tau_{i-1}^2 = \frac{1}{\eta} \sum_{i=k}^{T-1} \rho_{i+1}\tau_{i-1} \leq \frac{\tau}{\eta} \sum_{i=k}^{T-1} \rho_{i+1} \leq \frac{\tau}{\eta} \frac{\rho_{k+1}(1-\eta^{T-k}\tau^{T-k})}{1-\eta\tau} \leq \frac{\tau\rho_{k+1}}{\eta(1-\eta\tau)}$; ④ holds by setting $\bar{\eta} = \gamma\eta$.
Similarly, we can bound

$$
\begin{aligned}
\sum_{k=0}^{T-1} \bar{\Pi}_k' &= \sum_{k=0}^{T-1} \frac{\tau_{k-1}\rho_k\eta(\bar{\eta}-\eta)^2}{c_1^2} \left[ \sum_{i=0}^{k-1} \frac{1}{\rho_{i+1}(1-\eta\tau_{i-1})(1+\lambda_i\eta)^2} \|\boldsymbol{w}_i\|^2 \right] \\
&\leq \frac{\tau\eta(\bar{\eta}-\eta)^2}{c_1^2(1-\eta\tau)} \sum_{k=0}^{T-1} \rho_k \left[ \sum_{i=0}^{k-1} \frac{1}{\rho_{i+1}} \|\boldsymbol{w}_i\|^2 \right] \\
&\leq \frac{\tau\eta(\bar{\eta}-\eta)^2}{c_1^2(1-\eta\tau)} \sum_{k=0}^{T-1} \frac{\|\boldsymbol{w}_k\|^2}{\rho_{k+1}} \left[ \sum_{i=k}^{T-1} \rho_i \right] \\
&\overset{①}{\leq} \frac{(\bar{\eta}-\eta)^2}{c_1^2(1-\eta\tau)^2} \sum_{k=0}^{T-1} \left[ \|\boldsymbol{w}_k\|^2 \right] \leq \frac{\eta^2\gamma^2(\gamma-1)^2}{c_1^2(1+\gamma)^2} \sum_{k=0}^{T-1} \left[ \|\boldsymbol{w}_k\|^2 \right] \leq \frac{\eta^2(\gamma-1)^2}{c_1^2} \sum_{k=0}^{T-1} \left[ \|\boldsymbol{w}_k\|^2 \right]
\end{aligned}
\tag{20}
$$

where ① holds since 1) $\rho_{k+1} = \eta\tau_{k-1}\rho_k \leq \eta\tau\rho_k$ and $\rho_1 = 1$ and 2) $\sum_{i=k}^{T-1} \rho_i \leq \frac{\rho_k(1-\eta^{T-k}\tau^{T-k})}{1-\eta\tau} \leq \frac{\rho_k}{1-\eta\tau}$ which together give $\frac{1}{\rho_{k+1}} \left[ \sum_{i=k}^{T-1} \rho_i \right] \leq \frac{1}{\rho_{k+1}} \frac{\rho_k}{1-\eta\tau} \leq \frac{1}{\eta\tau} \frac{1}{1-\eta\tau} \leq \frac{1}{\eta\tau(1-\eta\tau)}$. Therefore, we

have

$$\frac{1}{T}\sum_{k=0}^{T-1}\mathbb{E}\left[\|\nabla F_k(\boldsymbol{x}_k)\|^2 + \frac{1}{2}\|\boldsymbol{w}_k\|^2\right]$$

$$\leq \frac{2c_2(1+\lambda\eta)\Delta}{\eta T} + \frac{2c_2\beta_1\sigma^2(1+\lambda\eta)}{c_1 b}$$

$$+ \frac{4c_2\gamma^2\eta^2(1-\beta_1)^2 L^2(1+\lambda\eta)(1+a^2\gamma^3)}{c_1^3\beta_1^2 T}\sum_{k=0}^{T-1}\left[\|\boldsymbol{w}_{k-1}\|^2\right] + \frac{2c_2\eta^2 L(\gamma-1)^2}{c_1^3 T}\sum_{k=0}^{T-1}\left[\|\boldsymbol{w}_k\|^2\right]$$

$$\overset{①}{\leq} \frac{2c_2(1+\lambda\eta)\Delta}{\eta T} + \frac{2c_2\beta_1\sigma^2(1+\lambda\eta)}{c_1 b} + \frac{1}{4T}\sum_{k=0}^{T-1}\left[\|\boldsymbol{w}_k\|^2\right]$$

where ① holds since we choose proper $\eta$ and $\beta_1$ such that

$$\frac{4c_2\gamma^2\eta^2(1-\beta_1)^2 L^2(1+\lambda\eta)(1+a^2\gamma^3)}{c_1^3\beta_1^2} \leq \frac{1}{8}$$

$$\frac{2c_2\eta^2 L(\gamma-1)^2}{c_1^3} \leq \frac{1}{8} \tag{21}$$

where ① uses $\frac{\gamma^3(\gamma-1)^2(1+\lambda\tau)}{(1+\gamma)^5} \leq 1 + \gamma\tau = 1 + \gamma\eta(\gamma+1)$ and ② uses $\frac{\gamma^3(\gamma-1)^2}{(1+\gamma)^4} < \gamma$. Now we select $\eta$ and $\beta_1$ such that (21) holds:

$$\eta \leq \min\left(\frac{c_1^{1.5}\beta_1}{4\sqrt{2}c_2^{0.5}\gamma(1-\beta_1)L(1+\lambda\eta)^{0.5}(1+a^2\gamma^3)^{0.5}}, \frac{c_1^{1.5}}{4c_2^{0.5}L^{0.5}(\gamma-1)}\right)$$

So we arrive at

$$\frac{1}{T}\sum_{k=0}^{T-1}\mathbb{E}\left[\|\nabla F_k(\boldsymbol{x}_k)\|^2 + \frac{1}{4}\|\boldsymbol{w}_k\|^2\right] \leq \frac{2c_2(1+\lambda\eta)\Delta}{\eta T} + \frac{2c_2\beta_1(1+\lambda\eta)\sigma^2}{c_1 b} \overset{①}{\leq} \epsilon^2, \tag{22}$$

where we set $T \geq \frac{4c_2(1+\lambda\eta)\Delta}{\eta\epsilon^2}$ and $\beta_1 \leq \frac{c_1 b\epsilon^2}{4c_2(1+\lambda\eta)\sigma^2}$. This result directly bounds

$$\frac{1}{T}\sum_{k=0}^{T-1}\|\boldsymbol{s}_k * (\boldsymbol{x}_k - \boldsymbol{x}_{k+1})\|^2 = \frac{\eta^2}{T}\sum_{k=0}^{T-1}\frac{1}{(1+\lambda_k\eta)^2}\|\boldsymbol{m}_k + \lambda\boldsymbol{x}_k * \boldsymbol{s}_k\|^2$$

$$\leq \frac{\eta^2}{T}\sum_{k=0}^{T-1}\|\boldsymbol{w}_k\|^2 \leq \eta^2\epsilon^2.$$

Moreover, from Lemma 5, we have

$$\frac{1}{T}\sum_{k=0}^{T-1}\|\boldsymbol{y}_k - (1+\lambda_{k-1}\bar{\eta})\boldsymbol{x}_k\|^2 \overset{①}{\leq} \frac{1}{T}\sum_{k=0}^{T-1}\rho_k(\bar{\eta}-\eta)^2\sum_{i=0}^{k-1}\frac{1}{\rho_{i+1}(1-\eta\tau_{i-1})(1+\lambda_i\eta)^2}\left\|\frac{\boldsymbol{w}_i}{\boldsymbol{s}_i}\right\|^2 \overset{③}{=} \frac{1}{T}\sum_{k=0}^{T-1}\Pi_k'',$$

$$\frac{1}{T}\sum_{k=0}^{T-1}\|\boldsymbol{z}_k - \boldsymbol{x}_k\|^2 \overset{①}{\leq} \frac{1}{T}\sum_{k=0}^{T-1}\tau_{k-1}\rho_k\eta(\bar{\eta}-\eta)^2\sum_{i=0}^{k-1}\frac{1}{\rho_{i+1}(1-\eta\tau_{i-1})(1+\lambda_i\eta)^2}\left\|\frac{\boldsymbol{w}_i}{\boldsymbol{s}_i}\right\|^2 \overset{②}{=} \frac{1}{T}\sum_{k=0}^{T-1}\Pi_k',$$

$$\frac{1}{T}\sum_{k=0}^{T-1}\|\boldsymbol{z}_{k+1} - \boldsymbol{z}_k\|^2 \overset{①}{\leq} \frac{1}{T}\sum_{k=0}^{T-1}\left[\frac{2\bar{\eta}^2}{(1+\lambda_k\eta)^2} + 2\rho_{k+1}\bar{\eta}^2(\bar{\eta}-\eta)^2\tau_k^2(1+\lambda_k\eta)^2\sum_{i=0}^{k}\frac{1}{\rho_{i+1}(1-\eta\tau_{i-1})(1+\lambda_i\eta)^2}\right]\left\|\frac{\boldsymbol{w}_k}{\boldsymbol{s}_k}\right\|^2$$

$$\overset{②}{\leq} \frac{1}{T}\sum_{k=0}^{T-1}\Pi_k$$

where $\rho_{k+1} = \eta\tau_{k-1}\rho_k$, $\rho_1 = 1$ and $\rho_0 = 0$. where ① holds by using Lemma 5; ② holds by using the definition in Eqn. (15); ③ holds by defining:

$$\Pi_k'' := \rho_k(\bar{\eta}-\eta)^2\sum_{i=0}^{k-1}\frac{1}{\rho_{i+1}(1-\eta\tau_{i-1})(1+\lambda_i\eta)^2}\left\|\frac{\boldsymbol{w}_i}{\boldsymbol{s}_i}\right\|^2.$$

Now remaining task is to upper bound $\frac{1}{T}\sum_{k=0}^{T-1}\Pi_k''$, $\frac{1}{T}\sum_{k=0}^{T-1}\Pi_k$ and $\frac{1}{T}\sum_{k=0}^{T-1}\Pi_k'$. Here we first bound $\frac{1}{T}\sum_{k=0}^{T-1}\Pi_k''$ by using almost the same proof in Eqn. (20):

$$
\begin{aligned}
\frac{1}{T}\sum_{k=0}^{T-1}\Pi_k'' &\overset{①}{\leq} \sum_{k=0}^{T-1}\frac{\rho_k(\bar{\eta}-\eta)^2}{c_1^2 T}\left[\sum_{i=0}^{k-1}\frac{1}{\rho_{i+1}(1-\eta\tau_{i-1})(1+\lambda_i\eta)^2}\|\boldsymbol{w}_i\|^2\right]\\
&\leq \frac{(\bar{\eta}-\eta)^2}{c_1^2(1-\eta\tau)T}\sum_{k=0}^{T-1}\rho_k\left[\sum_{i=0}^{k-1}\frac{1}{\rho_{i+1}}\|\boldsymbol{w}_i\|^2\right]\leq \frac{(\bar{\eta}-\eta)^2}{c_1^2(1-\eta\tau)T}\sum_{k=0}^{T-1}\frac{\|\boldsymbol{w}_k\|^2}{\rho_{k+1}}\left[\sum_{i=k}^{T-1}\rho_i\right]\\
&\overset{②}{\leq} \frac{(\bar{\eta}-\eta)^2}{c_1^2\eta\tau(1-\eta\tau)^2 T}\sum_{k=0}^{T-1}\left[\|\boldsymbol{w}_k\|^2\right]\leq \frac{\eta^2\gamma^2(\gamma-1)^2}{c_1^2(1+\gamma)T}\sum_{k=0}^{T-1}\left[\|\boldsymbol{w}_k\|^2\right]\leq \frac{\eta^2\gamma(\gamma-1)^2}{c_1^2 T}\sum_{k=0}^{T-1}\left[\|\boldsymbol{w}_k\|^2\right]\\
&\overset{③}{\leq} \frac{c_1\gamma}{16c_2 L}4\epsilon^2 = \frac{c_1\gamma\epsilon^2}{4c_2 L}
\end{aligned}
$$
(23)

where ① holds since $\frac{1}{1-\eta\tau_{i-1}}=\frac{\eta+\bar{\eta}+\lambda_{i-1}\bar{\eta}\eta}{\bar{\eta}+\lambda_{i-1}\bar{\eta}\eta}=1+\frac{\eta}{\bar{\eta}+\lambda_{i-1}\bar{\eta}\eta}\leq 1+\frac{\eta}{\bar{\eta}}=\frac{1}{1-\eta\tau}$ whose minimum is at $\lambda_{i-1}=0$ and $\tau=\frac{1}{\eta+\bar{\eta}}$; ② holds since 1) $\rho_{k+1}=\eta\tau_{k-1}\rho_k\leq\eta\tau\rho_k$ and $\rho_1=1$ and 2) $\sum_{i=k}^{T-1}\rho_i\leq\frac{\rho_k(1-\eta^{T-k}\tau^{T-k})}{1-\eta\tau}\leq\frac{\rho_k}{1-\eta\tau}$ which together give $\frac{1}{\rho_{k+1}}\left[\sum_{i=k}^{T-1}\rho_i\right]\leq\frac{1}{\rho_{k+1}}\frac{\rho_k}{1-\eta\tau}\leq\frac{1}{\eta\tau}\frac{1}{1-\eta\tau}\leq\frac{1}{\eta\tau(1-\eta\tau)}$; ③ holds by using 1) $\frac{1}{T}\sum_{k=0}^{T-1}\mathbb{E}\|\boldsymbol{w}_k\|^2\leq 4\epsilon^2$ in Eqn. (22); 2) we use the results in Eqn. (21) to obtain

$$
\frac{\eta^2\gamma(\gamma-1)^2}{c_1^2}\leq\frac{\gamma(\gamma-1)^2}{c_1^2}\frac{c_1^3}{16c_2 L(\gamma-1)^2}\leq\frac{c_1\gamma}{16c_2 L}.
$$

From the bound in Eqn. (16) and the following bound on $\frac{1}{T}\sum_{k=0}^{T-1}\bar{\Pi}_k$ and $\frac{1}{T}\sum_{k=0}^{T-1}\bar{\Pi}_k'$, we have

$$
\begin{aligned}
\frac{1}{T}\sum_{k=0}^{T-1}\Pi_k &\leq \frac{1}{T}\sum_{k=0}^{T-1}\bar{\Pi}_k\leq\frac{2\gamma^2\eta^2}{c_1^2 T}\left[1+a^2\gamma^3\right]\sum_{k=0}^{T-1}\mathbb{E}\left[\|\boldsymbol{w}_k\|^2\right]\overset{①}{\leq}\frac{c_1\beta_1^2\epsilon^2}{4c_2(1-\beta_1)^2 L^2(1+\lambda\eta)}\\
\frac{1}{T}\sum_{k=0}^{T-1}\Pi_k' &\leq \frac{1}{T}\sum_{k=0}^{T-1}\bar{\Pi}_k'\leq\frac{\eta^2(\gamma-1)^2}{c_1^2 T}\sum_{k=0}^{T-1}\mathbb{E}\left[\|\boldsymbol{w}_k\|^2\right]\overset{①}{\leq}\frac{c_1\epsilon^2}{4c_2 L}
\end{aligned}
$$

where ① holds, since 1) $\frac{1}{T}\sum_{k=0}^{T-1}\mathbb{E}\|\boldsymbol{w}_k\|^2\leq 4\epsilon^2$; 2) we use the results in Eqn. (21) to obtain

$$
\begin{aligned}
\frac{2\gamma^2\eta^2}{c_1^2}\left[1+a^2\gamma^3\right]&\leq\frac{2\gamma^2}{c_1^2}\left[1+a^2\gamma^3\right]\frac{c_1^3\beta_1^2}{32c_2\gamma^2(1-\beta_1)^2 L^2(1+\lambda\eta)(1+a^2\gamma^3)}\leq\frac{c_1\beta_1^2}{16c_2(1-\beta_1)^2 L^2(1+\lambda\eta)}\\
\frac{\eta^2(\gamma-1)^2}{c_1^2}&\leq\frac{(\gamma-1)^2}{c_1^2}\frac{c_1^3}{16c_2 L(\gamma-1)^2}\leq\frac{c_1}{16c_2 L}
\end{aligned}
$$

Therefore, we have

$$
\begin{aligned}
\frac{1}{T}\sum_{k=0}^{T-1}\mathbb{E}\left\|\boldsymbol{y}_k-(1+\lambda_k\bar{\eta})\boldsymbol{x}_k\right\|^2 &\leq\frac{c_1\gamma\epsilon^2}{4c_2 L},\\
\frac{1}{T}\sum_{k=0}^{T-1}\mathbb{E}\left\|\boldsymbol{z}_k-\boldsymbol{x}_k\right\|^2 &\leq\frac{c_1\epsilon^2}{4c_2 L},\\
\frac{1}{T}\sum_{k=0}^{T-1}\mathbb{E}\left\|\boldsymbol{z}_{k+1}-\boldsymbol{z}_k\right\|^2 &\leq\frac{c_1\beta_1^2\epsilon^2}{4c_2(1-\beta_1)^2 L^2(1+\lambda\eta)}.
\end{aligned}
$$

Besides, we have

$$
\begin{aligned}
\frac{1}{T}\sum_{k=0}^{T-1}\mathbb{E}\left[\|\boldsymbol{m}_k - \nabla F(\boldsymbol{x}_k)\|^2\right] &\leq \frac{1}{T}\sum_{k=0}^{T-1}\mathbb{E}\left[\|\boldsymbol{m}_k + \lambda_k\boldsymbol{x}_k * \boldsymbol{s}_k - \nabla F(\boldsymbol{x}_k) - \lambda_k\boldsymbol{x}_k * \boldsymbol{s}_k\|^2\right] \\
&\leq \frac{2}{T}\sum_{k=0}^{T-1}\mathbb{E}\left[\|\boldsymbol{m}_k + \lambda_k\boldsymbol{x}_k * \boldsymbol{s}_k\|^2 + \|\nabla F(\boldsymbol{x}_k) + \lambda_k\boldsymbol{x}_k * \boldsymbol{s}_k\|^2\right] \\
&= \frac{2}{T}\sum_{k=0}^{T-1}\mathbb{E}\left[\|\boldsymbol{m}_k + \lambda_k\boldsymbol{x}_k * \boldsymbol{s}_k\|^2 + \|\nabla F_k(\boldsymbol{x}_k)\|^2\right] \\
&\overset{①}{\leq} 2\left[\epsilon^2 + \frac{3}{4} \times 4\epsilon^2\right] \leq 8\epsilon^2.
\end{aligned}
$$

where in ① we use $\boldsymbol{w}_k = \boldsymbol{m}_k + \lambda_k\boldsymbol{x}_k * \boldsymbol{s}_k$. In this way, we have

$$
\begin{aligned}
\frac{1}{T}\sum_{k=0}^{T-1}\mathbb{E}\left[\|\boldsymbol{m}_k - \nabla F(\boldsymbol{z}_k)\|^2\right] &\leq \frac{2}{T}\sum_{k=0}^{T-1}\mathbb{E}\left[\|\boldsymbol{m}_k - \nabla F(\boldsymbol{x}_k)\|^2 + \|\nabla F(\boldsymbol{x}_k) - \nabla F(\boldsymbol{z}_k)\|^2\right] \\
&\leq 16\epsilon^2 + \frac{2L^2}{T}\sum_{k=0}^{T-1}\mathbb{E}\left[\|\boldsymbol{x}_k - \boldsymbol{z}_k\|^2\right] \\
&\leq 16\epsilon^2 + \frac{c_1 L\epsilon^2}{2c_2} = \frac{(c_1 L + 32c_2)}{2c_2}\epsilon^2.
\end{aligned}
$$

For all hyper-parameters, we put their constrains together:

$$
\beta_1 \leq \frac{c_1 b\epsilon^2}{4c_2(1+\lambda\eta)\sigma^2} = \mathcal{O}\left(\frac{c_1 b\epsilon^2}{c_2\sigma^2}\right),
$$

where $c_1 = \nu^{0.5} \leq \|\boldsymbol{s}_k\|_\infty \leq \left(c_\infty^2 + \nu\right)^{0.5} = c_2$.

For $\eta$, it should satisfy

$$
\eta \leq \min\left(\frac{c_1^{1.5}\beta_1}{4\sqrt{2}c_2^{0.5}\gamma(1-\beta_1)L(1+\lambda\eta)^{0.5}(1+a^2\gamma^3)^{0.5}}, \frac{c_1^{1.5}}{4c_2^{0.5}L^{0.5}(\gamma-1)}, \frac{c_1^2(1+\lambda\eta)}{2c_2(L+\lambda c_1)}\right)
$$

Considering $\lambda\eta << 1$, $\frac{1+\lambda\eta}{1+(1-\mu)\lambda\eta} = a \leq \frac{1}{1-\mu}$, $\mu$ is a constant, and $c_1 = \nu^{0.5} << 1$, then we have

$$
\begin{aligned}
\eta \leq& \mathcal{O}\left(\min\left(\frac{c_1^{1.5}\beta_1}{c_2^{0.5}\gamma^{2.5}L}, \frac{c_1^{1.5}}{c_2^{0.5}\gamma L^{0.5}}, \frac{c_1^2}{c_2 L}\right)\right) \\
=& \mathcal{O}\left(\min\left(\frac{c_1^{2.5}b\epsilon^2}{c_2^{1.5}\gamma^{2.5}\sigma^2 L}, \frac{c_1^{1.5}}{c_2^{0.5}\gamma L^{0.5}}, \frac{c_1^2}{c_2 L}\right)\right) = \mathcal{O}\left(\frac{c_1^{2.5}b\epsilon^2}{c_2^{1.5}\gamma^{2.5}\sigma^2 L}\right)
\end{aligned}
$$

where $\nu$ is often much smaller than one, and $\beta_1$ is very small. For $T$, we have

$$
\begin{aligned}
T \geq& \frac{4c_2(1+\lambda\eta)\Delta}{\eta\epsilon^2} = \mathcal{O}\left(\frac{c_2\Delta}{\epsilon^2}\frac{c_2^{1.5}\gamma^{2.5}\sigma^2 L}{c_1^{2.5}b\epsilon^2}\right) \\
=& \mathcal{O}\left(\frac{c_2^{2.5}\gamma^{2.5}\sigma^2 L\Delta}{c_1^{2.5}b\epsilon^4}\right) = \mathcal{O}\left(\frac{c_2^{2.5}\gamma^{2.5}\sigma^2 L\Delta}{\nu^{1.25}b\epsilon^4}\right).
\end{aligned}
$$

Now we compute the stochastic gradient complexity. For $T$ iterations, the complexity is

$$
\mathcal{O}(Tb) = \mathcal{O}\left(\frac{c_2^{2.5}\gamma^{2.5}\sigma^2 L\Delta}{\nu^{1.25}\epsilon^4}\right).
$$

The proof is completed.

$\square$

# F PROOFS OF THEOREM 2

*Proof.* Recall our definition $F_k(\boldsymbol{\theta}_k) = F(\boldsymbol{\theta}) + \frac{\lambda_k}{2}\|\boldsymbol{\theta}\|_2^2 = \mathbb{E}_{\boldsymbol{\zeta}}[f(\boldsymbol{\theta};\boldsymbol{\zeta})] + \frac{\lambda_k}{2}\|\boldsymbol{\theta}\|_2^2$ in the (13). By setting $\beta_1' = 1 - \beta_1$, then we have $\|\boldsymbol{m}_k\|_\infty \leq c_\infty$ by using Lemma 3 (see Appendix D). Also we define

$$\boldsymbol{w}_k := \boldsymbol{m}_k + \lambda\boldsymbol{x}_k, \qquad \boldsymbol{x}_{k+1} - \boldsymbol{x}_k = -\frac{\eta_k}{1+\lambda\eta_k}(\boldsymbol{m}_k + \lambda\boldsymbol{x}_k) = -\frac{\eta_k}{1+\lambda\eta_k}\boldsymbol{w}_k.$$

Note in the following, we set all $\lambda_k = \lambda$. By using the smoothness of $f(\boldsymbol{\theta};\boldsymbol{\zeta})$, we can obtain

$$F_{k+1}(\boldsymbol{x}_{k+1})$$

$$\leq F(\boldsymbol{x}_k) + \langle \nabla F(\boldsymbol{x}_k), \boldsymbol{x}_{k+1} - \boldsymbol{x}_k \rangle + \frac{L}{2}\|\boldsymbol{x}_{k+1} - \boldsymbol{x}_k\|^2 + \frac{\lambda}{2}\|\boldsymbol{x}_{k+1}\|^2$$

$$\overset{①}{\leq} F(\boldsymbol{x}_k) + \frac{\lambda}{2}\|\boldsymbol{x}_k\|^2 + \langle \nabla F(\boldsymbol{x}_k) + \lambda\boldsymbol{x}_k, \boldsymbol{x}_{k+1} - \boldsymbol{x}_k \rangle + \frac{L}{2}\|\boldsymbol{x}_{k+1} - \boldsymbol{x}_k\|^2 + \frac{\lambda}{2}\|\boldsymbol{x}_{k+1} - \boldsymbol{x}_k\|^2$$

$$= F_k(\boldsymbol{x}_k) - \frac{\eta_k}{1+\lambda\eta_k}\langle \nabla F(\boldsymbol{x}_k) + \lambda\boldsymbol{x}_k, \boldsymbol{w}_k \rangle + \frac{L\eta_k^2}{2(1+\lambda\eta_k)^2}\|\boldsymbol{w}_k\|^2 + \frac{\lambda\eta_k^2}{2(1+\lambda\eta_k)^2}\|\boldsymbol{w}_k\|^2$$

$$= F_k(\boldsymbol{x}_k) + \frac{1}{2}\left\|\sqrt{\frac{\eta_k}{(1+\lambda\eta_k)}}\left(\nabla F(\boldsymbol{x}_k) + \lambda\boldsymbol{x}_k - \boldsymbol{w}_k\right)\right\|^2 - \frac{1}{2}\left\|\sqrt{\frac{\eta_k}{(1+\lambda\eta_k)}}\left(\nabla F(\boldsymbol{x}_k) + \lambda\boldsymbol{x}_k\right)\right\|^2$$

$$- \frac{1}{2}\left\|\sqrt{\frac{\eta_k}{(1+\lambda\eta_k)}}\boldsymbol{w}_k\right\|^2 + \frac{L\eta_k^2}{2(1+\lambda\eta_k)^2}\|\boldsymbol{w}_k\|^2 + \frac{\lambda\eta_k^2}{2(1+\lambda\eta_k)^2}\|\boldsymbol{w}_k\|^2$$

$$\overset{②}{\leq} F_k(\boldsymbol{x}_k) + \frac{\eta_k}{2(1+\lambda\eta_k)}\|\nabla F(\boldsymbol{x}_k) - \boldsymbol{m}_k\|^2 - \frac{\eta_k}{2(1+\lambda\eta_k)}\|\nabla F_k(\boldsymbol{x}_k)\|^2$$

$$- \frac{\eta_k}{2(1+\lambda\eta_k)}\left[1 - \frac{L\eta_k}{(1+\lambda\eta_k)} - \frac{\lambda\eta_k}{(1+\lambda\eta_k)}\right]\|\boldsymbol{w}_k\|^2$$

$$\overset{③}{\leq} F_k(\boldsymbol{x}_k) + \frac{\eta_k}{2(1+\lambda\eta_k)}\|\nabla F(\boldsymbol{x}_k) - \boldsymbol{m}_k\|^2 - \frac{\eta_k}{2(1+\lambda\eta_k)}\|\nabla F_k(\boldsymbol{x}_k)\|^2 - \frac{\eta_k}{4(1+\lambda\eta_k)}\|\boldsymbol{w}_k\|^2,$$

where ① holds because

$$\|\boldsymbol{x}_{k+1}\|_{\boldsymbol{s}_k}^2 = \|\boldsymbol{x}_k\|_{\boldsymbol{s}_k}^2 + \|\boldsymbol{x}_{k+1} - \boldsymbol{x}_k\|_{\boldsymbol{s}_k}^2 + 2\langle \boldsymbol{x}_{k+1} - \boldsymbol{x}_k, \boldsymbol{x}_k \rangle_{\boldsymbol{s}_k}.$$

② holds, because

$$\boldsymbol{w}_k := \boldsymbol{m}_k + \lambda\boldsymbol{x}_k, \qquad \boldsymbol{x}_{k+1} - \boldsymbol{x}_k = -\frac{\eta_k}{1+\lambda\eta_k}(\boldsymbol{m}_k + \lambda\boldsymbol{x}_k) = -\frac{\eta_k}{1+\lambda\eta_k}\boldsymbol{w}_k.$$

④ holds, since we set $\eta_k \leq \frac{c_1^2(1+\lambda\eta_k)}{2c_2(L+\lambda c_1)}$ such that $\frac{c_2 L\eta_k}{c_1^2(1+\lambda\eta_k)} + \frac{c_2\lambda\eta_k}{c_1(1+\lambda\eta_k)} \leq \frac{1}{2}$.

Then in the following, we can directly follow the proof of Theorem 1. This is because the only difference between accelerated SGD and AdamW is that SGD has no the second-order moment $\boldsymbol{v}_k$, while AdamW has. By let $\boldsymbol{s}_k = \mathbf{1}$ in accelerated AdamW and setting $\beta_1' = 1 - \beta_1$ in accelerated SGD, then they share the exact the same updating rules. So after setting $\beta_1' = 1 - \beta_1$ in accelerated SGD, to follow the proofs of Theorem 1, we only need to verify whether the auxiliary lemmas and the proof process of Theorem 1 hold for $\boldsymbol{s}_k = \mathbf{1}$. This is the true case. Please check our auxiliary lemmas, including Lemma $3 \sim 6$, and the proof process of Theorem 1. Consider $\boldsymbol{s}_k = \mathbf{1}$ in accelerated SGD, we have $c_1 := 1 \leq \|\boldsymbol{s}_k\|_\infty \leq c_2 := 1$.

In this way, by setting $\bar{\eta}_k = \gamma\eta_k, \gamma > 1, \eta_k = \eta \leq \mathcal{O}\left(\frac{b\epsilon^2}{c^{1.5}\gamma^{2.5}\sigma^2 L}\right), \beta_1 \leq \mathcal{O}\left(\frac{b\epsilon^2}{c\sigma^2}\right), \beta_1' = 1 - \beta_1, \lambda_k = \lambda, \lambda_0 = 0$, after $T = \mathcal{O}\left(\frac{\Delta\sigma^2 L}{b\epsilon^4}\right)$ iterations with minibatch size $b$ and $\Delta = F(\boldsymbol{x}_0) - F(\boldsymbol{x}_\star)$, the sequence $\{(\boldsymbol{x}_k, \boldsymbol{z}_k)\}_{k=0}^T$ generated by accelerated SGD satisfies the following four properties.

**a)** The gradient $\nabla F_k(\boldsymbol{x}_k)$ of the sequence $\{\boldsymbol{x}_k\}_{k=0}^T$ can be upper bounded by

$$\frac{1}{T}\sum_{k=0}^{T-1}\mathbb{E}\left[\|\nabla F_k(\boldsymbol{x}_k)\|_2^2 + \frac{1}{4}\|\boldsymbol{m}_k + \lambda_k\boldsymbol{x}_k\|_2^2\right] \leq \epsilon^2.$$

**b)** The gradient moment $\boldsymbol{m}_k$ can well estimate the full gradient $\nabla F(\boldsymbol{x}_k)$ and $\nabla F(\boldsymbol{z}_k)$:

$$\frac{1}{T}\sum_{k=0}^{T-1}\max\left\{\mathbb{E}\|\boldsymbol{m}_k - \nabla F(\boldsymbol{x}_k)\|_2^2, \mathbb{E}\|\boldsymbol{m}_k - \nabla F(\boldsymbol{z}_k)\|_2^2\right\} \leq \left(16 + \frac{1}{2}L\right)\epsilon^2.$$

**c)** The sequence $\{\boldsymbol{x}_k, \boldsymbol{z}_k\}$ satisfies

$$\frac{1}{T}\sum_{k=0}^{T-1}\left\{\mathbb{E}\|\boldsymbol{x}_k-\boldsymbol{x}_{k+1}\|^2, \mathbb{E}\|\boldsymbol{z}_{k+1}-\boldsymbol{z}_k\|_2^2, \mathbb{E}\|\boldsymbol{z}_k-\boldsymbol{x}_k\|_2^2\right\}\le\left\{4\eta^2\epsilon^2, \frac{\beta_1^2\epsilon^2}{4(1-\beta_1)^3L^2}, \frac{\epsilon^2}{4L}\right\}.$$

**d)** The total stochastic gradient complexity to achieve the above three properties is $\mathcal{O}\big(\frac{c_\infty^{2.5}\Delta\sigma^2 L}{\nu^{1.25}\epsilon^4}\big)$. The proof is completed. $\qquad\square$

## G   PROOFS OF AUXILIARY LEMMAS

### G.1   PROOF OF LEMMA 3

*Proof.* To begin with, we assume that $\forall t \le k$, it holds

$$\|\boldsymbol{m}_t\|_\infty \le c_\infty, \qquad \|\boldsymbol{v}_t+\nu\|_\infty \le c_\infty+\nu$$

Then we consider the case where $t = k+1$ as follows

$$\|\boldsymbol{m}_{k+1}\|_\infty = \|(1-\beta_1)\boldsymbol{m}_k+\beta_1\boldsymbol{g}_k\|_\infty \le (1-\beta_1)\|\boldsymbol{m}_k\|_\infty+\beta_1\|\boldsymbol{g}_k\|_\infty \le c_\infty,$$

$$\|\boldsymbol{v}_{k+1}\|_\infty = \|(1-\beta_2)\boldsymbol{v}_k+\beta_2\boldsymbol{g}_k^2\|_\infty \le (1-\beta_2)\|\boldsymbol{v}_k\|_\infty+\beta_2\|\boldsymbol{g}_k^2\|_\infty \le c_\infty^2.$$

Then we derive the second results as follows:

$$\left\|\sqrt{\frac{\boldsymbol{v}_k+\nu}{\boldsymbol{v}_{k+1}+\nu}}\right\|_\infty = \left\|\sqrt{1+\frac{\boldsymbol{v}_k-\boldsymbol{v}_{k+1}}{\boldsymbol{v}_{k+1}+\nu}}\right\|_\infty = \left\|\sqrt{1+\frac{\beta_2(\boldsymbol{v}_k-\boldsymbol{g}_k^2)}{\boldsymbol{v}_{k+1}+\nu}}\right\|_\infty.$$

Therefore, we have

$$1-\frac{\beta_2 c_\infty^2}{2(c_{s,\infty}^2+\nu)} < \sqrt{1-\frac{\beta_2 c_\infty^2}{c_{s,\infty}^2+\nu}} \le \left\|\sqrt{\frac{\boldsymbol{v}_k+\nu}{\boldsymbol{v}_{k+1}+\nu}}\right\|_\infty \le \sqrt{1+\frac{\beta_2 c_\infty^2}{c_{s,\infty}^2+\nu}} < 1+\frac{\beta_2 c_\infty^2}{2(c_{s,\infty}^2+\nu)}.$$

We complete the proof. $\qquad\square$

### G.2   PROOF OF LEMMA 5

*Proof.* To begin with, we have

$$\boldsymbol{y}_{k+1}-(1+\lambda_k\bar{\eta}_k)\boldsymbol{x}_{k+1}$$

$$=\boldsymbol{z}_k-\bar{\eta}_k\frac{\boldsymbol{m}_k}{\boldsymbol{s}_k}-\frac{1+\lambda_k\bar{\eta}_k}{1+\lambda_k\eta_k}\left(\boldsymbol{x}_k-\eta_k\frac{\boldsymbol{m}_k}{\boldsymbol{s}_k}\right)$$

$$=\bar{\eta}_{k-1}\tau_{k-1}\boldsymbol{x}_k+\eta_{k-1}\tau_{k-1}\boldsymbol{y}_k-\bar{\eta}_k\frac{\boldsymbol{m}_k}{\boldsymbol{s}_k}-\frac{1+\lambda_k\bar{\eta}_k}{1+\lambda_k\eta_k}\left(\boldsymbol{x}_k-\eta_k\frac{\boldsymbol{m}_k}{\boldsymbol{s}_k}\right)$$

$$=\eta_{k-1}\tau_{k-1}\left(\boldsymbol{y}_k-(1+\lambda_k\bar{\eta}_{k-1})\boldsymbol{x}_k\right)-\left(\bar{\eta}_k-\frac{1+\lambda_k\bar{\eta}_{k-1}}{1+\lambda_k\eta_{k-1}}\eta_k\right)\frac{\boldsymbol{m}_k}{\boldsymbol{s}_k}+\frac{\lambda_k(\eta_k-\bar{\eta}_k)}{1+\lambda_k\eta_k}\boldsymbol{x}_k$$

$$\overset{①}{=}\eta_{k-1}\tau_{k-1}\left(\boldsymbol{y}_k-(1+\lambda_k\bar{\eta}_{k-1})\boldsymbol{x}_k\right)-\left(\bar{\eta}_k-\frac{1+\lambda_k\bar{\eta}_{k-1}}{1+\lambda_k\eta_{k-1}}\eta_k\right)\frac{\boldsymbol{w}_k-\lambda_k\sqrt{\boldsymbol{v}_k}}{\boldsymbol{s}_k}+\frac{\lambda_k(\eta_k-\bar{\eta}_k)}{1+\lambda_k\eta_k}\boldsymbol{x}_k$$

$$=\eta_{k-1}\tau_{k-1}\left(\boldsymbol{y}_k-(1+\lambda_k\bar{\eta}_{k-1})\boldsymbol{x}_k\right)-\left(\bar{\eta}_k-\frac{1+\lambda_k\bar{\eta}_{k-1}}{1+\lambda_k\eta_{k-1}}\eta_k\right)\frac{\boldsymbol{w}_k}{\boldsymbol{s}_k}$$

$$+\left(\lambda_k\bar{\eta}_k-\frac{1+\lambda_k\bar{\eta}_{k-1}}{1+\lambda_k\eta_{k-1}}\lambda_k\eta_k+\frac{\lambda_k(\eta_k-\bar{\eta}_k)}{1+\lambda_k\eta_k}\right)\boldsymbol{x}_k$$

$$\overset{②}{=}\eta\tau_{k-1}\left(\boldsymbol{y}_k-(1+\lambda_k\bar{\eta})\boldsymbol{x}_k\right)-\frac{\bar{\eta}-\eta}{1+\lambda_k\eta}\frac{\boldsymbol{w}_k}{\boldsymbol{s}_k}$$

where ① holds since $\boldsymbol{w}_k := \boldsymbol{m}_k+\lambda_k\boldsymbol{x}_k*\boldsymbol{s}_k$; ② holds since we set all $\eta_k = \eta$ and $\bar{\eta}_k = \bar{\eta}$ which gives $\tau_k = \tau = \frac{1}{\eta+\bar{\eta}+\lambda_k\eta\bar{\eta}}$. Therefore, by defining $\rho_{k+1} = \eta\tau_{k-1}\rho_k$, $\rho_1 = 1$ and $\rho_0 = 0$, then we have

$$\frac{\boldsymbol{y}_{k+1}-(1+\lambda_k\bar{\eta})\boldsymbol{x}_{k+1}}{\rho_{k+1}}=\frac{\boldsymbol{y}_k-(1+\lambda_k\bar{\eta})\boldsymbol{x}_k}{\rho_k}-\frac{1}{\rho_{k+1}}\frac{\bar{\eta}-\eta}{1+\lambda_k\eta}\frac{\boldsymbol{w}_k}{\boldsymbol{s}_k}\ (k\ge 1)$$

For $k = 0$, we have

$$
\begin{aligned}
\boldsymbol{y}_1 - (1 + \lambda_0\bar{\eta})\boldsymbol{x}_1 =& \boldsymbol{z}_0 - \bar{\eta}\frac{\boldsymbol{m}_0}{\boldsymbol{s}_0} - \frac{1 + \lambda_0\bar{\eta}}{1 + \lambda_0\eta}\left(\boldsymbol{x}_0 - \eta\frac{\boldsymbol{m}_0}{\boldsymbol{s}_0}\right) \\
=& \boldsymbol{z}_0 - \bar{\eta}\frac{\boldsymbol{w}_0 - \lambda_0\boldsymbol{s}_0 * \boldsymbol{x}_0}{\boldsymbol{s}_0} - \frac{1 + \lambda_0\bar{\eta}}{1 + \lambda_0\eta}\left(\boldsymbol{x}_0 - \eta\frac{\boldsymbol{w}_0 - \lambda_0\boldsymbol{s}_0 * \boldsymbol{x}_0}{\boldsymbol{s}_0}\right) \\
=& \boldsymbol{z}_0 - \boldsymbol{x}_0 - \frac{\bar{\eta} - \eta}{1 + \lambda_0\eta}\frac{\boldsymbol{w}_0}{\boldsymbol{s}_0}
\end{aligned}
$$

In this way, one can obtain

$$
\begin{aligned}
\frac{\boldsymbol{y}_{k+1} - (1 + \lambda_k\bar{\eta})\boldsymbol{x}_{k+1}}{\rho_{k+1}} =& \boldsymbol{z}_0 - \boldsymbol{x}_0 - \frac{\bar{\eta} - \eta}{1 + \lambda_0\eta}\frac{\boldsymbol{w}_0}{\boldsymbol{s}_0} - \sum_{i=1}^{k}\frac{1}{\rho_{i+1}}\frac{\bar{\eta} - \eta}{1 + \lambda_i\eta}\frac{\boldsymbol{w}_i}{\boldsymbol{s}_i} \\
=& -\sum_{i=0}^{k}\frac{1}{\rho_{i+1}}\frac{\bar{\eta} - \eta}{1 + \lambda_i\eta}\frac{\boldsymbol{w}_i}{\boldsymbol{s}_i}
\end{aligned}
$$

where ① hold since $\boldsymbol{z}_0 = \boldsymbol{x}_0$ and $\rho_1 = 1$. Then we can upper bound

$$
\begin{aligned}
\left\|\frac{\boldsymbol{y}_{k+1} - (1 + \lambda_k\bar{\eta})\boldsymbol{x}_{k+1}}{\rho_{k+1}}\right\|^2 =& \left\|\sum_{i=0}^{k}\frac{\rho_{k+1}(1 - \eta\tau_{i-1})}{\rho_{i+1}}\frac{\bar{\eta} - \eta}{\rho_{k+1}(1 - \eta\tau_{i-1})(1 + \lambda_i\eta)}\frac{\boldsymbol{w}_i}{\boldsymbol{s}_i}\right\|^2 \\
\overset{①}{\leq}& \sum_{i=0}^{k}\frac{\rho_{k+1}(1 - \eta\tau_{i-1})}{\rho_{i+1}}\frac{(\bar{\eta} - \eta)^2}{\rho_{k+1}^2(1 - \eta\tau_{i-1})^2(1 + \lambda_i\eta)^2}\left\|\frac{\boldsymbol{w}_i}{\boldsymbol{s}_i}\right\|^2 \\
=& \frac{(\bar{\eta} - \eta)^2}{\rho_{k+1}}\sum_{i=0}^{k}\frac{1}{\rho_{i+1}(1 - \eta\tau_{i-1})(1 + \lambda_i\eta)^2}\left\|\frac{\boldsymbol{w}_i}{\boldsymbol{s}_i}\right\|^2
\end{aligned}
$$

where ① holds since 1) $\sum_{i=0}^{k}\frac{1 - \eta\tau_{i-1}}{\rho_{i+1}} = \sum_{i=0}^{k}\left(\frac{1}{\rho_{i+1}} - \frac{1}{\rho_i}\right) = \frac{1}{\rho_{k+1}}$, and 2) Jensen' inequality. Therefore, we have

$$
\|\boldsymbol{y}_{k+1} - (1 + \lambda_k\bar{\eta})\boldsymbol{x}_{k+1}\|^2 \leq \rho_{k+1}(\bar{\eta} - \eta)^2\sum_{i=0}^{k}\frac{1}{\rho_{i+1}(1 - \eta\tau_{i-1})(1 + \lambda_i\eta)^2}\left\|\frac{\boldsymbol{w}_i}{\boldsymbol{s}_i}\right\|^2.
$$

Moreover, we can also bound

$$
\begin{aligned}
\|\boldsymbol{z}_{k+1} - \boldsymbol{x}_{k+1}\|^2 =& \|\bar{\eta}\tau_k\boldsymbol{x}_{k+1} + \eta\tau_k\boldsymbol{y}_{k+1} - \boldsymbol{x}_{k+1}\|^2 \\
=& \eta\tau_k\|\boldsymbol{y}_{k+1} - (1 + \lambda_k\bar{\eta})\boldsymbol{x}_{k+1}\|^2 \\
\leq& \tau_k\rho_{k+1}\eta(\bar{\eta} - \eta)^2\sum_{i=0}^{k}\frac{1}{\rho_{i+1}(1 - \eta\tau_{i-1})(1 + \lambda_i\eta)^2}\left\|\frac{\boldsymbol{w}_i}{\boldsymbol{s}_i}\right\|^2.
\end{aligned}
$$

On the other hand, we have

$$
\begin{aligned}
\|\boldsymbol{z}_{k+1} - \boldsymbol{z}_k\| =& \|\bar{\eta}\tau_k\boldsymbol{x}_{k+1} + \eta\tau_k\boldsymbol{y}_{k+1} - \boldsymbol{z}_k\| \\
\overset{①}{=}& \left\|\bar{\eta}\tau_k\boldsymbol{x}_{k+1} + \eta\tau_k\boldsymbol{y}_{k+1} - \boldsymbol{y}_{k+1} - \bar{\eta}\frac{\boldsymbol{m}_k}{\boldsymbol{s}_k}\right\| \\
=& \left\|\bar{\eta}\tau_k\boldsymbol{x}_{k+1} + \eta\tau_k\boldsymbol{y}_{k+1} - \boldsymbol{y}_{k+1} - \bar{\eta}\frac{\boldsymbol{w}_k - \lambda_k\boldsymbol{x}_k * \boldsymbol{s}_k}{\boldsymbol{s}_k}\right\| \\
=& \left\|\bar{\eta}\tau_k\boldsymbol{x}_{k+1} + \eta\tau_k\boldsymbol{y}_{k+1} - \boldsymbol{y}_{k+1} - \bar{\eta}\frac{\boldsymbol{w}_k}{\boldsymbol{s}_k} + \bar{\eta}\lambda_k\boldsymbol{x}_k\right\| \\
\overset{②}{=}& \left\|(\bar{\eta}\tau_k + \bar{\eta}\lambda_k)\boldsymbol{x}_{k+1} - (1 - \eta\tau_k)\boldsymbol{y}_{k+1} - \frac{\bar{\eta}}{1 + \lambda_k\eta}\frac{\boldsymbol{w}_k}{\boldsymbol{s}_k}\right\| \\
\overset{③}{=}& \left\|\bar{\eta}\tau_k(1 + \lambda_k\eta)\left((1 + \lambda_k\bar{\eta})\boldsymbol{x}_{k+1} - \boldsymbol{y}_{k+1}\right) - \frac{\bar{\eta}}{1 + \lambda_k\eta}\frac{\boldsymbol{w}_k}{\boldsymbol{s}_k}\right\| \\
\leq& \bar{\eta}\tau_k(1 + \lambda_k\eta)\|(1 + \lambda_k\bar{\eta})\boldsymbol{x}_{k+1} - \boldsymbol{y}_{k+1}\| + \frac{\bar{\eta}}{1 + \lambda_k\eta}\left\|\frac{\boldsymbol{w}_k}{\boldsymbol{s}_k}\right\|
\end{aligned}
$$

where ① we plug in $\boldsymbol{y}_{k+1} = \boldsymbol{z}_k - \bar{\eta}_k \frac{\boldsymbol{m}_k}{\boldsymbol{s}_k}$; in ② we plug in $\boldsymbol{x}_{k+1} = \frac{1}{1+\lambda_k \eta} \left( \boldsymbol{x}_k - \eta \frac{\boldsymbol{m}_k}{\boldsymbol{s}_k} \right) = \frac{1}{1+\lambda_k \eta} \left( \boldsymbol{x}_k - \eta \frac{\boldsymbol{w}_k - \lambda_k \boldsymbol{x}_k * \boldsymbol{s}_k}{\boldsymbol{s}_k} \right) = \boldsymbol{x}_k - \frac{\eta}{1+\lambda_k \eta} \frac{\boldsymbol{w}_k}{\boldsymbol{s}_k}$; and ③ we have $\bar{\eta} \tau_k + \bar{\eta} \lambda_k = \bar{\eta} \tau_k (1 + \bar{\eta} \lambda_k)(1 + \eta \lambda_k)$ and $(1 - \eta \tau_k) = \bar{\eta} \tau_k (1 + \eta \lambda_k)$. Then we can upper bound

$$\|\boldsymbol{z}_{k+1} - \boldsymbol{z}_k\|^2 \leq 2\bar{\eta}^2 \tau_k^2 (1 + \lambda_k \eta)^2 \|(1 + \lambda_k \bar{\eta})\boldsymbol{x}_{k+1} - \boldsymbol{y}_{k+1}\|^2 + \frac{2\bar{\eta}^2}{(1 + \lambda_k \eta)^2} \left\| \frac{\boldsymbol{w}_k}{\boldsymbol{s}_k} \right\|^2$$

$$\leq \frac{2\bar{\eta}^2}{(1 + \lambda_k \eta)^2} \left\| \frac{\boldsymbol{w}_k}{\boldsymbol{s}_k} \right\|^2 + 2\rho_{k+1} \bar{\eta}^2 (\bar{\eta} - \eta)^2 \tau_k^2 (1 + \lambda_k \eta)^2 \sum_{i=0}^{k} \frac{1}{\rho_{i+1}(1 - \eta \tau_{i-1})(1 + \lambda_i \eta)^2} \left\| \frac{\boldsymbol{w}_i}{\boldsymbol{s}_i} \right\|^2$$

The proof is completed. $\qquad \square$

## G.3 PROOF OF LEMMA 6

*Proof.* From Lemma 4, we have

$$\mathbb{E} \left[ \|\boldsymbol{m}_k - \nabla F(\boldsymbol{z}_k)\|^2 \right]$$

$$\leq (1 - \beta_{1,k}) \mathbb{E} \left[ \|\boldsymbol{m}_{k-1} - \nabla F(\boldsymbol{z}_{k-1})\|^2 \right] + \frac{(1 - \beta_{1,k})^2 L^2}{\beta_{1,k}} \mathbb{E} \left[ \|\boldsymbol{z}_k - \boldsymbol{z}_{k-1}\|^2 \right] + \frac{\beta_{1,k}^2 \sigma^2}{b}$$

$$\overset{①}{\leq} (1 - \beta_{1,k}) \mathbb{E} \left[ \|\boldsymbol{m}_{k-1} - \nabla F(\boldsymbol{z}_{k-1})\|^2 \right] + \frac{\Pi_k (1 - \beta_{1,k})^2 L^2}{\beta_{1,k}} + \frac{\beta_{1,k}^2 \sigma^2}{b}$$

where in ①, we use the results in Lemma 5 that

$$\|\boldsymbol{z}_k - \boldsymbol{z}_{k-1}\|^2 \leq \Pi_k$$

with

$$\Pi_k := \frac{2\bar{\eta}^2}{(1 + \lambda_{k-1} \eta)^2} \left\| \frac{\boldsymbol{w}_{k-1}}{\boldsymbol{s}_{k-1}} \right\|^2 + 2\rho_k \bar{\eta}^2 (\bar{\eta} - \eta)^2 \tau_{k-1}^2 (1 + \lambda_{k-1} \eta)^2 \sum_{i=0}^{k-1} \frac{1}{\rho_{i+1}(1 - \eta \tau_{i-1})(1 + \lambda_i \eta)^2} \left\| \frac{\boldsymbol{w}_i}{\boldsymbol{s}_i} \right\|^2.$$

Then we have

$$\mathbb{E} \left[ \|\boldsymbol{m}_k - \nabla F(\boldsymbol{x}_k)\|^2 \right] \leq 2\mathbb{E} \left[ \|\boldsymbol{m}_k - \nabla F(\boldsymbol{z}_k)\|^2 \right] + 2\mathbb{E} \left[ \|\nabla F(\boldsymbol{z}_k) - \nabla F(\boldsymbol{x}_k)\|^2 \right]$$

$$\leq 2\mathbb{E} \left[ \|\boldsymbol{m}_k - \nabla F(\boldsymbol{z}_k)\|^2 \right] + 2L\mathbb{E} \left[ \|\boldsymbol{z}_k - \boldsymbol{x}_k\|^2 \right]$$

$$\overset{①}{\leq} 2(1 - \beta_{1,k}) \mathbb{E} \left[ \|\boldsymbol{m}_{k-1} - \nabla F(\boldsymbol{z}_{k-1})\|^2 \right] + \frac{2\Pi_k (1 - \beta_{1,k})^2 L^2}{\beta_{1,k}} + \frac{2\beta_{1,k}^2 \sigma^2}{b} + 2L\Pi_k',$$

where in ①, we use the results in Lemma 4 that

$$\mathbb{E} \left[ \|\boldsymbol{m}_k - \nabla F(\boldsymbol{z}_k)\|^2 \right]$$

$$\leq (1 - \beta_{1,k}) \mathbb{E} \left[ \|\boldsymbol{m}_{k-1} - \nabla F(\boldsymbol{z}_{k-1})\|^2 \right] + \frac{(1 - \beta_{1,k})^2 L^2}{\beta_1} \mathbb{E} \left[ \|\boldsymbol{z}_k - \boldsymbol{z}_{k-1}\|^2 \right] + \frac{\beta_{1,k}^2 \sigma^2}{b}.$$

and also the results in Lemma 5 that

$$\|\boldsymbol{z}_k - \boldsymbol{x}_k\|^2 \leq \Pi_k' := \tau_{k-1} \rho_k \eta (\bar{\eta} - \eta)^2 \sum_{i=0}^{k-1} \frac{1}{\rho_{i+1}(1 - \eta \tau_{i-1})(1 + \lambda_i \eta)^2} \left\| \frac{\boldsymbol{w}_i}{\boldsymbol{s}_i} \right\|^2.$$

The proof is completed. $\qquad \square$

