# OpenReview forum: "Win: Weight-Decay-Integrated Nesterov Acceleration for Adaptive Gradient Algorithms"
_ICLR.cc/2023/Conference — ICLR 2023 notable top 5%_

### Official Review · Reviewer_FLq8 · 2022-10-18

**Confidence:** 3
**Correctness:** 4
**Technical Novelty And Significance:** 3
**Empirical Novelty And Significance:** 3
**Recommendation:** 8

**Clarity, Quality, Novelty And Reproducibility:**

The paper is quite clear. The novelty is a bit less clear, as the combination is used before, and the derivation from proximal point methods is well known.

**Strength And Weaknesses:**

The technique presented is well motivated theoretically, and is straightforward to implement. The experiments are quite diverse and show promising improvements across the board.

The organization of the paper could be a bit better --- right now the authors present the fully general technique and then specialize it to various algorithms, it might be better to present the technique for various algorithms one at a time, starting with SGD, which is the simplest algorithm.

The combination of weight decay and Nesterov momentum has been applied in other optimizers already (LARS, Shampoo, several others).

**Summary Of The Paper:**

The paper provides a new analysis of combining weight decay and Nesterov momentum and applying it to several standard first order optimizers (SGD, Adam, AdamW, LAMB). The authors prove convergence, and then show that the new acceleration provides significant improvements on a variety of training tasks.

**Summary Of The Review:**

The paper presents an analysis of combining weight decay with nesterov momentum. The technique has been applied to several different optimizers, and is easy to implement in practice. The experimental results show significant gains across a variety of ML tasks.

---

> ### Author Response · Authors · 2022-11-10
> **Response to Reviewer FLq8**
>
> Thank you for the insightful and very positive comments! In the following,   we provide our point-by-point response and hope our response helps address your concerns.  We also look forward to the subsequent discussion which may further help solve the current issues.
>
> **(1)** For **organization in this work**, we introduce Win for adaptive algorithms by using AdamW and Adam as examples in Sec. 3.1, and extend this acceleration technique to LAMB and SGD.  We first simultaneously use AdamW and Adam instead of SGD as examples to introduce Win because of the following two reasons.  **a)** Adaptive algorithms are much more popular than  SGD on the transformer (e.g. ViT, Swin, BERT, GPT, T5, and their variants) which is the currently most popular and effective architecture because of its superior performance and compatibility for both vision and language data at least in the research community. Indeed, for many SoTA CNN-type architectures (e.g. ResNet, RegNetY, ResNeXt, SENet, EfficientNet), as mentioned in Sec. 5.1, Wightman et al. (2021) have largely pushed their performance limits by using  LAMB adaptive method and stronger data augmentation, e.g. +4.3\% top-1 accuracy on  ResNet50. So first introducing adaptive algorithms detailedly, e.g. AdamW and Adam,  could attract more readers and help practicers better which also accords with our target that develops a practical acceleration for network training. **b)** Extending the acceleration framework of AdamW in Sec. 3.1 Accelerating SGD is quite direct and simple, but not vice versa, which can avoid too many repetitions and gives a more compact introduction. This is because the only algorithmic difference
> between SGD and AdamW is that SGD has no second-order moment $\boldsymbol{v}_k$ while AdamW has, and thus we can use the acceleration framework of AdamW in Sec. 3.1 to accelerate SGD by setting $\boldsymbol{v}_k=\boldsymbol{s}_k=\boldsymbol{1}\in\mathbb{R}^{d}$ in Eqn. (4), (5), and (6) to obtain WIN-accelerated SGD.  On the contrary, one cannot directly extend WIN-accelerated SGD to AdamW due to the more complicated algorithmic steps in AdamW.  But we also agree that introducing SGD first may be more simple for readers, and will consider this very good suggestion in the final version which allows us to use an extra page and thus gives more imagination for the organization.
>
> We simultaneously introduce Win into AdamW and Adam because of two benefits. AdamW and Adam have only one difference, i.e. the regularization parameter $\lambda>0$ in AdamW and  $\lambda=0$ in Adam.  So the first benefit is that it could give a more compact introduction due to the page limitation.  Another more important benefit is that simultaneously introducing AdamW and Adam allows readers to compare and observe the derivation differences more easily when plugging Win into an algorithm with weight decay (AdamW) and the one without weight decay (Adam), which may be more heuristic and could motivate or introduce other accelerations for adaptive algorithms.
>
> **(2) The aforementioned  LARS and Shampoo actually do not use Nesterov acceleration at least in their official papers**.  In the official Algorithm 1 of  LARS (arXiv:1708.03888v3),  given the stochastic gradient $\boldsymbol{g}_k$, LARS first updates the first-order moment $\boldsymbol{v} _ { k + 1 } = \beta_1 \boldsymbol{v} _ {k} + \gamma \boldsymbol{g} _ k'$, and then updates the parameter $\boldsymbol{w} _ {k+1} = \boldsymbol{w} _ {k} - \boldsymbol{v} _ {k+1}$, where $\boldsymbol{g} _ k'= \frac{||\boldsymbol{w} _ {k}|| _ 2}{||\boldsymbol{g} _ k|| _ 2 + \lambda  ||\boldsymbol{w} _ {k}|| _ 2} (\boldsymbol{g} _ k + \lambda \boldsymbol{w} _ {k})$.   So LARS indeed scales $\boldsymbol{g} _ k + \lambda \boldsymbol{w} _ {k}$ to the same magnitude of the network weight $\boldsymbol{w} _ {k}$ (the same spirit in LAMB), and then uses the standard heavy ball acceleration to compute first-order moment $\boldsymbol{v} _ {k+1}$ which differs from the Nesterov acceleration as shown in Eqn. (10).  For Shampoo, it is a quasi-second-order method and indeed does not use heavy-ball or  Nesterov acceleration. In its official Algorithms 1 and 2 (see arXiv:1802.09568v2), given the stochastic gradient $ \boldsymbol{G}_k\in\mathbb{R}^{m\times n}$, Shampoo first updates the  preconditioners $\boldsymbol{L} _ {k+1} = \boldsymbol{L} _ {k} + \boldsymbol{G} _ k \boldsymbol{G} _ k ^ {\top}$ and $\boldsymbol{R} _ {k+1} = \boldsymbol{R} _ {k} + \boldsymbol{G} _ k ^ {\top} \boldsymbol{G} _ k$, and  then updates the parameter $\boldsymbol{w} _ {k+1} = \boldsymbol{w} _ {k} - \eta \boldsymbol{L} _ {k+1} ^ {-1/4}  \boldsymbol{G} _ {k} \boldsymbol{R} _ {k+1} ^ {-1/4}$. So Shampoo shares a similar precondition idea with some second-order methods which precondition the problem according to the geometry curvature of the loss objective.

---

> > ### Comment · Reviewer_FLq8 · 2022-11-11
> > **Nesterov and Weight Decay in prior work**
> >
> > Weight Decay and Nesterov in LARS and Shampoo implementations:
> >
> > https://optax.readthedocs.io/en/latest/api.html#optax.lars
> >
> > optax.lars(learning_rate, weight_decay=0.0, weight_decay_mask=True, trust_coefficient=0.001, eps=0.0, trust_ratio_mask=True, momentum=0.9, nesterov=False)
> >
> > Parameters
> >
> > weight_decay (float) – Strength of the weight decay regularization.
> >
> > nesterov (bool) – Whether to use Nesterov momentum.
> >
> > https://github.com/google-research/google-research/blob/master/scalable_shampoo/jax/shampoo.py
> >
> > class _ShampooHyperParams:
> >
> >   """Shampoo hyperparameters."""
> >
> >   \# Weight decay parameter for regularization.
> >
> >   weight_decay: float
> >
> >   \# Nesterov momentum
> >
> >   nesterov: bool

---

> > > ### Author Response · Authors · 2022-11-15
> > > **Response to Weight Decay and Nesterov in LARS and Shampoo implementations**
> > >
> > > Many thanks for your code. From these codes, one can find that **both accelerated LARS and Shampoo  use the same vanilla  Nesterov acceleration in SGD, and their Nesterov acceleration differs from Win-acceleration** which is discussed in Sec. 3.2. Moreover, these implementations indeed are not proposed  in their official papers as  discussed above.
> > >
> > > Here **we use Nesterov-accelerated LARS (LARS-M) in the provided code as an example for explanation**.
> > > For more clarity, we first recall  Nesterov-accelerated SGD (SGD-M for short) whose updating steps are
> > >
> > > $
> > > \boldsymbol{m} _ {k} =   \beta _ {1} \boldsymbol{m} _ {k-1} + \beta _ {1}' (\boldsymbol{g} _ {k} + \lambda \boldsymbol{x} _ {k} ), \quad
> > > \boldsymbol{x} _ {k+1} =  (1 - \lambda \eta _ {k}) \boldsymbol{x} _ {k} - \eta _ {k} (\boldsymbol{g} _ {k}    + \beta _ {2} \boldsymbol{m} _
> > >  {k}). \qquad \qquad (1)
> > > $
> > >
> > > where $\boldsymbol{g} _ {k}$ denotes the stochastic gradient at the $k$-the iteration, $\beta_{1}\in[0,1]$ and $\beta_{1}'\in[0,1]$.
> > >
> > > LARS-M in the provided code shares very similar updating rules with SGD but  scales $\boldsymbol{g} _ k + \lambda \boldsymbol{w} _ {k}$ to the same magnitude of the network weight $\boldsymbol{w}_{k}$ for convergence acceleration:
> > >
> > > $
> > > 		\boldsymbol{m} _ {k} =   \beta _ {1} \boldsymbol{m} _ {k-1} + \beta _ {1}'\boldsymbol{g} _ k', \quad
> > > 		\boldsymbol{x} _ {k+1} = (1-\lambda \eta _ {k})\boldsymbol{x} _ {k} - \eta _ {k} (\boldsymbol{g} _ {k}'   + \beta _ {2} \boldsymbol{m} _ {k}). \qquad \qquad \qquad \qquad (2)
> > > $
> > >
> > > where $\boldsymbol{g} _ {k}'= \frac{ \| \boldsymbol{w} _ {k} \| _ 2 }{ \| \boldsymbol{g} _ k \| _ 2 + \lambda  \| \boldsymbol{w} _ {k} \|  _ 2} (\boldsymbol{g} _ k + \lambda \boldsymbol{w} _ {k} )$.
> > >
> > > By comparing Eqn. (1) and (2), both updating equations can be uniformly formulated into   Eqn. (2) where $\boldsymbol{g} _ {k}'= \frac{ \| \boldsymbol{w} _ {k} \| _ 2 }{ \| \boldsymbol{g} _ k \| _ 2 + \lambda  \| \boldsymbol{w} _ {k} \|  _ 2} (\boldsymbol{g} _ k + \lambda \boldsymbol{w} _ {k} )$ in LARS-M and $\boldsymbol{g} _ {k}' = \boldsymbol{g} _ k $ in SGD. This actually means that LARS-M and SGD-M both use the vanilla Nesterov acceleration which differs from the acceleration in Win as discussed in Sec. 3.2.  Please find more detailed comparison in Sec. 3.2.  **For the provided  Shampoo code**, it also uses the same Nesterov acceleration techniques in SGD-M and LARS-M, and thus it also distinguishes from the one used in Win.

---

### Official Review · Reviewer_kZaA · 2022-10-21

**Confidence:** 4
**Correctness:** 3
**Technical Novelty And Significance:** 4
**Empirical Novelty And Significance:** 4
**Recommendation:** 8

**Clarity, Quality, Novelty And Reproducibility:**

My main concern with this work is the clarity of Sections 1 and 3. I believe the clarity of writing and presentation for this work could be improved, which would make the technique much easier to understanding (i.e., try to be less verbose in writing, organize the ideas better with a clear structure, and be more clear in your explanations). If the authors focus upon improving this presentation, I believe the work could be very well received and impactful.

Novelty/quality/reproducibility seem good. It would be nice for authors to provide a reference implementation of Win (e.g., in PyTorch) for the different optimizers they consider.

**Strength And Weaknesses:**

Strength:
* The incorporation of the PPM regularization term with a recently-proposed formulation of nesterov-type acceleration is novel and seems to be useful.
* There proposed Win acceleration can be added to several different types of optimizers (e.g., Adam, SGD, AdamW, LAMB).
* The authors provide convergence guarantees in the non convex setting for their acceleration technique using both Adam and AdamW. These guarantees improve upon prior work to the best of my knowledge.
* Experiments are performed on a wide variety of architectures (CNNs, ViTs, LSTMs, transformers). X-Win improves upon baseline optimization techniques in nearly all cases.
* The proposed acceleration technique seems to reduce the need for hyperparameter tuning.

Weakness:
* The discussion of adaptive optimization techniques is lacking. It is widely recognized that adaptive methods suffer poor generalization relative to SGDM in certain settings (see “The marginal value of adaptive gradient methods in machine learning” and “Towards theoretically understanding why sgd generalizes better than adam in deep learning.”). This has led to research in alternative optimizers like AMSGrad, Yogi, M-SVAG, etc. The authors mention AdamW, but do not mention the rest of this work or sufficiently address the generalization ailments of adaptive methods. I think because of these issues, claiming that optimizers like Adam/AdamW are “default optimizers to train DNNs” is somewhat problematic/incorrect. SGDM is still the default choice for most computer vision tasks, where event recent adaptive algorithms like AdamW achieve inferior performance.
* The description of the proposed method (especially in the introduction) is lacking in clarity. I believe the paper could be greatly improved if the authors spend time to improve the clarity of these initial explanations of their technique.
* No discussion of how the computational complexity of the proposed method compares to the other methods (maybe I missed it?). You need to state this comparison, do we incur extra cost by adding Win?

Small stuff:
* Small writing errors: beginning of Section 3 (“accelerating and also stabilizing optimization”, “At below”), end of section 3.1 (shouldn’t “bias correlation” be “bias correction”?).
* Putting AdamW, Adam, and LAMB variants all into Algorithm 1 is a bit confusing, though I’m not sure if there is a better way to present this.
* It is not completely clear to me from your description how Win+LAMB compares to vanilla LAMB. Making this comparison a bit more directly would be helpful in my opinion.
* It is a bit weird that Table 1 appears after Table 2 in the writing.
* For the loss plots (Fig. 1), I’m not sure whether your method actually converges faster, or if it simply converges to a better loss. In some cases, the convergence does appear to be faster, but in others it converges at (seemingly) the same to a better loss. It would be helpful to provide a similar plot with some of the other baseline optimizers (e.g., PAdam, RAdam, Adabelief, etc.) if it is does not require too much extra effort.

**Summary Of The Paper:**

This work focuses on adaptive gradient optimization algorithms and attempts to provide practical insights for speeding up convergence with such methods. In particular, the authors proposed the Win method to increase the speed of convergence for adaptive optimization techniques. In particular, this technique incorporates a dynamic regularizer into the loss that is inspired by proximal point methods, which improves the convexity of the optimization problem. In addition to providing a convergence proof of the Win technique, the authors first implement this technique for Adam/AdamW, then extend the implementation to LAMB (i.e., technique for optimization with larger step sizes) and SGD. The authors provides convergence guarantees for this acceleration technique within the non convex settings, which matches known lower bounds and slightly improves upon the bounds achieved for baseline algorithms like AdaBelief and RMSProp. In experiments, the Win acceleration is shown to provide a very tangible benefit across a suite of different deep learning architectures, including CNNs, LSTMs, and transformers for both vision and language.

**Summary Of The Review:**

To begin, I want to emphasize that this review reflects my current opinion of the work. I am completely open to discussion with other reviewers/authors, and my final score will be mostly based upon this discussion.

My initial thoughts are as follows:
* The idea in this paper is unique and (in my opinion) useful to the community.
* The empirical (and seemingly theoretical, though I'm less familiar with Adam-style analysis) support for this technique is very strong.
* The presentation quality of the work is quite poor (especially Sec. 1, 3).

I believe that if the authors improve the clarity of their discussion/presentation of the technique, the impact of this work will significantly increase. I do not necessarily believe poor writing is a reason for rejection, especially given the strong empirical/theoretical support for Win. However, I truly believe that spending time with improving the writing of this paper (and fixing the other details I mention in the body of my review) will increase this work's potential significantly.

I thank the authors for their interesting work, and I look forward to the subsequent discussion!

---

> ### Author Response · Authors · 2022-11-10
> **Part 2 of Response to Reviewer kZaA**
>
> **(3)** For the **Win description** especially in the introduction, we will try our best to improve the clarity of Win.  At present, to introduce vanilla Nesterov acceleration, the derivations of AdamW, Adam, LAMB, and SGD, and their corresponding theoretical results within the limited space (9 pages), we have to put some similar techniques together, e.g. the derivations and theoretical analysis of AdamW and Adam. Due to the same reason, we also need to briefly introduce some contents, e.g. vanilla LAMB and SGD, and extend the acceleration derivations of AdamW to LAMB and SGD without much more detail. So the manuscript might read slightly brief without many details but a bit dense especially in definitions and inline formulations.
>
>
> We very much appreciate your suggestions about further improving the readability and will try our best to accordingly revise the paper, e.g. by introducing the initial explanations of Win more intuitively, giving more details of LAMB, SGD, and their acceleration frameworks, and also reducing inline formulations.   Moreover,  the final version allows us to use an extra page and thus gives more imagination for the explanation and discussion according to your suggestion and also other reviewers' suggestions.
>
>
> **(4)** We provide the **code of Win-accelerated AdamW in the supplementary materials** whose implementation is based on the widely used  Timm codebase (https://github.com/rwightman/pytorch-image-models). If you want to test, please follow the provided ``readme.md". For other optimizers, we are cleaning up and will release it online along with our final paper version.
>
> **Small Stuffs:**
>
>
> **(1)** For **the typos**, we have fixed them in the new version per your suggestion.
>
> **(2) For  AdamW, Adam, and LAMB  in Algorithm 1, they only differ from the 5-th step and share all remaining steps.** This is the reason why we put them into one algorithm which can avoid too many repetitions and saves space. To make it more clear, in revision we will separate the inline formulation of the  5-th step for all optimizers into two cases which are formulated as follows:
>
> $$\text{Option for AdamW and Adam}:  \boldsymbol{u} _ k =
> \frac{\boldsymbol{m} _ k}{\sqrt{\boldsymbol{v} _ {k} + \nu}}
> $$
>
> $$
> \text{Option for LAMB}: \boldsymbol{u} _ k =
> \frac{|| \boldsymbol{x} _ {k} || _ 2}{||\boldsymbol{m} _ {k} / \sqrt{ \boldsymbol{v} _ {k} + \nu } || _ 2} \frac{\boldsymbol{m} _ {k}}{\sqrt{\boldsymbol{v} _ {k} + \nu}}.  \\
> $$
>
>
> We do not do it now, since there is no sufficient space to do it due to the limitation of 9 pages.
>
>
> **(3)** To **compare Win+LAMB with vanilla LAMB**, we add Appendix A of the new revision to summarize their algorithmic steps in Algorithms 2 and 3. By observing  Algorithms 2 and 3, one can easily find their differences. Please refer to Appendix A for more details.  We also mention this detailed comparison in Appendix A  when introducing LAMB and Win-accelerated LAMB.
>
> **(4)** For **the loss plots in Fig. 1**, one can observe that for all test cases, Win-accelerated methods achieve small losses than the corresponding non-accelerated counterparts, especially after about the 10-th epochs. This means that with the same training epochs, Win-accelerated methods actually have faster convergence behaviors, otherwise they cannot obtain smaller losses. The differences among these plots are that some Win-accelerated methods show much notable faster convergence behaviors, while some accelerated ones have relatively small convergence advantages.
>
> For other baseline optimizers (e.g., PAdam, RAdam, Adabelief),  we will try to reproduce their results and plot them in one figure. It should be mentioned that reproducing deep learning algorithms is actually hard and cannot be finished in a few days, since it depends on many factors, e.g. the environments (PyTorch/TensorFlow version, Timm version, random seed) and the hyper-parameters, e.g. learning rate, warm-up epochs, and algorithm-related parameters.  If we can reproduce their results, we will certainly include them in the loss plot.

---

> > ### Comment · Reviewer_kZaA · 2022-11-15
> > **Response to authors (Part 2)**
> >
> > Thank you for the clarifications.
> >
> > 3. Thank you for putting work into the writing. Again, this is not a reason for rejection. It is simply a suggestion, as I think improving the writing a little bit can make the paper even better.
> >
> > 4. Thank you for the link. Providing the code addresses this concern obviously :)
> >
> > Small stuff.
> > 1. Thank you
> > 2. Thank you for the explanation.
> > 3. Understood, I think these modifications will help.
> > 4.
> > * To me it seems like the convergence speed is the same for the first few epochs (i.e., curves are nearly identical), but the Win-based algorithms "keep going" longer than the baselines, leading to a lower final loss. I find this behavior pretty interesting! Indeed, this implies that Win actually converges faster, so the statement is correct.
> > * I was mostly referring to the fact that this initial period of similar convergence convergence speed. If you have any intuition for why the curves look like this, it would be great to provide. But, this is no big deal, I simply find it interesting.
> > * Totally understand the concerns with adding extra baselines. They would be good to add, but the paper already has extensive empirical results, so this is not a problem for me.
> >
> > Thanks to the authors for the discussion. It helped me to better understand the paper and clarify all of my remaining questions. To reflect this, I raise my score to an 8 (accept).  Again, I would suggest putting some extra writing effort into Sec 1/3 and fixing the small points that are mentioned. Either way, I thank the authors for their interesting work and willingness to address all of my questions. Congratulations on the awesome paper.

---

> ### Author Response · Authors · 2022-11-10
> **Part 1 of Response to Reviewer kZaA**
>
>  Thank you for your insightful and positive comments, and also for your very careful proofreading! In the following,   we provide our point-by-point response and hope our response helps address your concerns. We also look forward to the subsequent discussion which may further help solve the current issues.
>
> **(1) Discussion of adaptive optimization techniques.**  We would like to first clarify the claim that ``adaptive optimizers like Adam/AdamW become default optimizers in training DNNs", and then introduce a more precise and rigorous claim per your suggestion.
>
>  On the transformers, such as vision transformers (ViT, Swin and their variants) in the computer vision field and transformers (BERT, GPT, Transform-XL, T5, and their variants) in the natural language processing domain, adaptive gradient methods, e.g. Adam and AdamW, are indeed the default optimizers and are much more popular than SGD-M. For many SoTA CNN-type architectures (e.g. ResNet, RegNetY, ResNeXt, SENet, EfficientNet), as mentioned in Sec. 5.1, the recently proposed “A2 training recipe” in (Wightman et al., 2021) has largely pushed their performance limits by using stronger data augmentation and LAMB adaptive method,  e.g. +4.3\% top-1 accuracy on  ResNet50.   This is the main reason for our claim that adaptive optimizers become “default optimizers to train DNNs”.
>
>
>  But as you mentioned, before (Wightman et al., 2021),  SGD is more popular in  CNN training because of its better performance than adaptive methods, and many works have paid much effort and contributed a lot to improve the generalization performance of adaptive algorithms.  So according to your suggestion and the popularity of adaptive algorithms in current transformer and CNN architectures, to be more precise, we change our vanilla claim to a sentence-alike ``adaptive optimizers like Adam/AdamW become more popular in training DNNs, especially for transformers". We have already revised all the claims in the manuscript.
>
>
>  For other adaptive optimization works (e.g. theoretical works from Wilson et al. and Zhou et al., and practical optimizers, including AMSGrad, Yogi, MSVAG, etc.), we will discuss them in the related work of the final version which allows us to use an extra page, since at present, there is really no space to add new content. It should be mentioned that in the new version, we have compared Yogi,  MSVAG, and fromage (Bernstein et al., NeurIPS'20)  in our experimental section. Indeed, in the experiment, we do compare ours with several other adaptive optimizers, e.g.   AdaBound, Radam, Nadam, AdaBelief, and Padam.
>
>
> **(2)** On **extra overhead, our Win-accelerated method introduces extra negligible computational cost**. This is because **a)** as discussed in the second paragraph in the 4-th page, Win acceleration only adds one extra simple algorithmic step, i.e. the 7-th step $\boldsymbol{z} _ {k+1} = \bar{\eta} _ {k} \tau _ {k} \boldsymbol{x} _ {k+1} +  {\eta} _ {k} \tau _ {k} (\boldsymbol{y} _ {k+1} -  \bar{\eta} _ {k} \boldsymbol{u} _ {k})$ in Algorithm 1, into the vanilla optimizer; **b)**  since this extra step only linearly combines $\boldsymbol{x} _ {k+1} $, $\boldsymbol{y} _ {k+1}$ and $ \boldsymbol{u} _ k$, it is very simple and its cost can be negligible compared the existing network forward and back-propagation evaluation in the vanilla optimizer. Note,  network forward and back-propagation evaluation are much slower than this extra step.  Indeed, on our test models, e.g. ViT-small and ViT-base models, for a single iteration, AdamW-Win is only about  1.02$\times$ slower averagely than  AdamW, and indeed share almost the same speed. We have briefly introduced this discussion in the revision, and will detailedly introduce it in the final version.

---

> > ### Comment · Reviewer_kZaA · 2022-11-15
> > **Response to authors (Part 1)**
> >
> > Thank you for the clarifications
> >
> > 1. The explanation makes complete sense and actually provides me with extra insight into the current state of adaptive optimizers. I believe slightly adjusting the wording of these claims as you suggest is more than sufficient.
> >
> > 2. This again emphasizes the practicality of Win. I believe mentioning the negligible computational cost (as you suggest) will make this more clear, so this addresses my question.

---

> > > ### Author Response · Authors · 2022-11-16
> > > **Response to Reviewer kZaA**
> > >
> > > Many thanks. We are glad our response can help you better understand our work, and address your concerns. We will spend more effort and time to improve the writing, and also try our best to solve the issues mentioned. Thank you for your insightful and positive comments again!!

---

### Official Review · Reviewer_hKYG · 2022-11-01

**Confidence:** 4
**Clarity, Quality, Novelty And Reproducibility:** 1. Clarity and quality
**Correctness:** 4
**Technical Novelty And Significance:** 3
**Empirical Novelty And Significance:** 3
**Recommendation:** 8

**Strength And Weaknesses:**

Strength: The paper is clear and well written. It presents an acceleration algorithm and apply to Adam, AdamW, LAMB and SGD. It has both theretical analysis and experiments to support the claims. The idea introduced is nice.

Weaknesses:
1. Better to include more comparisons with other methods, such as the line search methods.
2. To consider the s_k norm seems not necessary. What if s_k is partial, like 1/2?


**Summary Of The Paper:**

This paper presents a weight decay integrated Nesterov acceleration algorithm for Adam, AdamW, LAMB and SGD by adopting the proximal point method. It shows the convergence analysis, and perform experiments to validate the results.

**Summary Of The Review:**

It is a very well written paper.

---

> ### Author Response · Authors · 2022-11-10
> **Part 2 of Response to Reviewer hKYG**
>
>
> **(2)** We provide the **code of Win-accelerated AdamW in the supplementary materials** whose implementation is based on the widely used  Timm codebase (https://github.com/rwightman/pytorch-image-models). If you want to test, please follow the provided ``readme.md". For other Win-accelerated optimizers, we are cleaning up and will release it online along with our final paper version.
>
>
> **(3)** For the question related to **the $s_k$ norm**, could you introduce it in more detail? Since there are several places that involve the $s_k$ norm. If you mean the $s_k$ norm when we extend Win acceleration framework of AdamW to  SGD, then we can directly set $\boldsymbol{v}_k = \boldsymbol{s}_k = \boldsymbol{1} \in \mathbb{R}^{d}$ in Eqn. (4), (5) and (6) of AdamW acceleration derivation to obtain WIN-accelerated SGD. This is because the only algorithmic difference
> between SGD and AdamW is that SGD has no second-order moment $\boldsymbol{v}_k$  while AdamW has. So SGD actually corresponds to the AdamW version with $\boldsymbol{v}_k = \boldsymbol{s}_k = \boldsymbol{1} \in \mathbb{R}^{d}$ in Eqn. (4), (5) and (6). If your question refers to other places,  please specifically point out the place so that we can explain it more clearly.  Thanks.

---

> ### Author Response · Authors · 2022-11-10
> **Part 1 of Response to Reviewer hKYG**
>
>  Thank you for the insightful and very positive comments! In the following,   we provide our point-by-point response and hope our response helps address your concerns. We also look forward to the subsequent discussion which may further help solve the current issues.
>
> **(1)** For  **baselines**, we have added  Yogi (Zaheer et al., NeurIPS'18),  MSVAG (Balles et al., ICML'18), and fromage (Bernstein  et al., NeurIPS'20) in our new revision. Specifically, Yogi and MSVAG respectively achieve 68.2\% and  66.0\% top-1 accuracy on ResNet18;  fromage and MSVAG respectively obtain 68.0 and 65.3 test perplexity on LSTM. These results are quoted from the official AdaBelief paper because of the same setting.  For comparison, please refer to Tables 1 and 5 in the revision.  Note, on the large-scale dataset (e.g. ImageNet), for other representative optimizers (e.g. AdaBound, Radam, and AdaBlief), they only test on ResNet18 and do not evaluate on other networks. In this way, it is hard for us to compare them on ResNet50/101, ViTs, PoolFormers, etc, due to the challenges for hyper-parameter tuning of these optimizers and our limited GPU resources. Note on ResNet18, we already compare all these baselines.
>
>
>  In practice, **line search methods** are not used in network training due to their high cost.  Taking the representative  Backtracking Line Search (BLS) in the optimization field as an example.  Let the loss object be $F(\boldsymbol{x}) =  \mathbb{E}_{\zeta \sim \mathcal{D}} [f(\boldsymbol{x},\zeta)]$ and the  search direction at the $k$-th training iteration be $\boldsymbol{d}_k$ (e.g. $\boldsymbol{d}_k=-\nabla F(\boldsymbol{x})/\|\nabla F(\boldsymbol{x})\|_2$ in BLS). Then at the $(k+1)$-th iteration,   BLS starts with a relatively large step size $\eta$, and repeatedly shrinks it by a factor $\alpha\in (0,1)$ until the Armijo–Goldstein condition $F(\boldsymbol{x}_k + \eta \boldsymbol{d}_k) \leq F(\boldsymbol{x}_k) + c  \eta \boldsymbol{d}_k^{\top} \nabla F(\boldsymbol{x}_k)$ is fulfilled where $c \in(0,1]$ is a constant.
>
> There are two important factors that lead to the high training cost of BLS.  **a)**  For each step size candidate $\eta$, one needs to perform the network forward for computing the loss $F(\boldsymbol{x}_k + \eta \boldsymbol{d}_k)$ which is actually very expensive, especially for modern networks that are of large size in most cases. What is worse, line search methods often require trying at least several step-size candidates to satisfy the Armijo–Goldstein condition, which linearly increases the network forward evaluation number and prohibits their applications in practice.  **b)** Conventional BLS actually needs to compute the full losses (including $F(\boldsymbol{x}_k + \eta \boldsymbol{d}_k)$  and $F(\boldsymbol{x}_k)$) and full gradient $\nabla F(\boldsymbol{x}_k)$ on all samples which are too expensive to use in practice. This is because modern networks are often large and thus are trained on a large-scale dataset, e.g. the most widely used ImageNet dataset of 1.26 million images. Other line search methods also suffer from these two similar issues. So in this work, we do not compare the vanilla BLS.
>
> To comparison, we instead implement a stochastic version of BLS by using minibatch loss to approximate the full losses $F(\boldsymbol{x}_k + \eta \boldsymbol{d}_k)$  and $F(\boldsymbol{x}_k)$, and also estimating the full gradient $\nabla F(\boldsymbol{x}_k)$  in search direction $\boldsymbol{d}_k=-\nabla F(\boldsymbol{x})/\|\nabla F(\boldsymbol{x})\|_2$ as the minibatch gradient.  Then we test it on SGD-M (i.e. SGD + Nesterov acceleration) on ResNet18 by setting $\eta=10 \eta_k'$ and $c=0.8$ where $\eta_k'$ is the vanilla learning rate in SGD-M. Following the conventional 90  training epoch setting, SGD-M  with  BLS only achieves 68.1\% which is much lower than 70.2\% of vanilla   SGD-M, and its overall training time for 90 epochs is about $4.4\times $  more than the training time of vanilla   SGD-M. Its inferior performance is possibly caused by its greedy strategy which always finds a lower loss and may impose restrictions on the exploration ability which is important to escape the local minima that widely exist in deep networks.  Due to its much inferior performance and also high cost, we do not include it in our manuscript, since this work focuses more on the practical aspect of efficient network training.

---

### Public Comment · ~Rohan_Anil1 · 2022-11-16
**Clarification on Nesterov + Adam results**

As pointed by Reviewer FLq8, adding nesterov acceleration to existing adaptive methods is standard: See comment: https://openreview.net/forum?id=CPdc77SQfQ5&noteId=7fpHqssTGr4 -- (note: improvements from nesterov with shampoo have been mentioned in https://arxiv.org/pdf/2002.09018.pdf Table 1 Pg 11 as well as in https://arxiv.org/abs/2209.05310)

There is an experimental confound when citing Nadam as the implementation in Pytorch (https://pytorch.org/docs/stable/generated/torch.optim.NAdam.html) is quite different from the implementation (vanilla) in JAX as mentioned in FLq8 -- It would be good to clarify which nesterov version is used in Table 1 Page 7.

For the experiments that are in Table 3 - A comparison against the vanilla nesterov + adam/lamb variants would be appreciated [1] -- This would establish if the extra memory overhead in WIN compared to vanilla nesterov is worth it.

[1] see: https://github.com/google/init2winit/blob/master/init2winit/optimizer_lib/optimizers.py#L190

---

> ### Author Response · Authors · 2022-11-18
> **Response to Rohan**
>
> Dear Rohan Anil,
>
> Many thanks for your insightful comments.  In the following, we provide our point-by-point response and hope our response helps address your concerns. We also look forward to the subsequent discussion which may further help solve the current issues.
>
> **(1)** Thanks for your clarification on  Nesterov acceleration on adaptive methods.  As we explained in the reply to Reviewer FLq8,  **accelerated Shampoo use the same vanilla  Nesterov acceleration in SGD, and their Nesterov acceleration differs from Win-acceleration which is discussed in Sec. 3.2**.   Besides, this implementation indeed is not proposed in the official Shampoo paper, but is very briefly mentioned in later works.  In the final version which allows us to use an extra page, we will discuss these related works for their big efforts on  Nesterov acceleration and adaptive methods, e.g. (arXiv:2002.09018v2) and (arXiv:2209.05310) as aforementioned.  We do not discuss this now, since there is really no space to add new content.
>
>
> **(2) For Nadam**, we use its official Pytorch implementation in our experiments.  Per your suggestion, we will mention this information in our experiment section.
>
> **(3)** Per your suggestion, we have implemented vanilla **Nesterov accelerated Adam** (Adam-M for short), and compared them with our Win-accelerated Adam on ResNet18 with ImageNet. Here we follow the Nesterov implementation in Shampoo (see https://github.com/google-research/google-research/blob/master/scalable_shampoo/pytorch/shampoo.py) to accelerate Adam.
>
> As shown in the following Table, one can observe that **Win-accelerated Adam still achieves higher classification performance than vanilla Nesterov-accelerated Adam (Adam-M)**.  In the experiments, we independently run both Adam-M and Win-accelerated Adam under four different stepsizes, and report their best accuracy. Indeed, this stepsize tuning also slightly improves Win-accelerated Adam from 68.8\% to 69.3\%.  Note, for other optimizers run by us, we already well-tune their hyper-parameters.  Moreover, as shown in Table 1  of vanilla submission  (also the following table), **Win-accelerated SGD also surpasses Nesterov accelerated SGD (SGD-M in table) and heavy-ball accelerated SGD (SGD-H)**. These results are consistent and both show the effectiveness of Win acceleration. We will include this in the revision.
>
> Table: ImageNet top-1 accuracy (\%) of ResNet18.  $*$ and $\dagger$  are respectively reported in (arXiv:1806.06763) and (arXiv:2010.07468).
> |Adam   | 66.5$^{\dagger}$ | SGD-H | 67.3 |
> | -------- | ----- | ------------- | ----------------- |
> |Adam-M   | 67.7  | SGD-M | 70.2$^*$ |
> |Adam-Win | 69.3 |	SGD-Win   | 70.7|
>
> Here we do not test on vision transformers in Table 3, since a) this extra question only leaves one day for us to do experiments;  b) in this short time, we can only finish the experiments on the small networks, e.g. ResNet18,  instead of vision transformers which is large and often require 3 days on 8 GPUs. Due to the same reason, we also cannot finish the experiments on LAMB. But in Tables 2 and 3 of vanilla submission, we also compare Win-accelerated SGD with vanilla Nesterov accelerated SGD denoted by SGD-M in the tables.  From the results, one can observe that **Win-accelerated SGD always achieves better performance than SGD-M on ResNet50, ResNet101, ViT-small, ViT-base, and PoolFormer-S12.** Please refer to the detailed comparison in Tables 2 and 3.

---

### Decision · Program_Chairs · 2023-01-20

**Decision:**

Accept: notable-top-5%

**Justification For Why Not Higher Score:**

N/A.

**Justification For Why Not Lower Score:**

Adaptive optimizers recently becomes more important in training large scale deep neural networks, particularly for transformers. The Nesterov acceleration combined with weight decay proposed in this paper shows tangible improvement over standard adaptive optimizers with a theoretical convergence support. So this paper can be a spotlight that may help the practice in speeding up training popular neural network architectures.

**Metareview: Summary, Strengths And Weaknesses:**

The paper proposes a new algorithm of combining weight decay and Nesterov acceleration, called WIN (Weight-decay Integrated Nesterov acceleration). It is applied to extend several standard first order optimizers (Adam, AdamW, LAMB and SGD). The authors provide convergence proof, and experimental evidences that the new acceleration provides significant improvements on a variety of training tasks, including CNNs, LSTMs, and transformers for both vision and language. The proposed idea is unique, with strong theoretical and empirical supports. All the reviewers unanimously accept the paper, so is the final decision.

**Note From Pc:**

if the above contains the word "oral" or "spotlight" please see: "oral" presentation means -> notable-top-5% and "spotlight" means -> notable-top-25%. As stated in our emails, we are disassociating presentation type from AC recommendations